# TGF-β blockade drives a transitional effector phenotype in T cells reversing SIV latency and decreasing SIV reservoirs in vivo

Jinhee Kim[1], Deepanwita Bose [2,11], Mariluz Araínga[2,11], Muhammad R. Haque[3], Christine M. Fennessey[4], Rachel A. Caddell[5], Yanique Thomas[3], Douglas E. Ferrell[2], Syed Ali[2], Emanuelle Grody[3,6], Yogesh Goyal [3,6,7], Claudia Cicala[8], James Arthos [8], Brandon F. Keele [4], Monica Vaccari [5,9], Ramon Lorenzo-Redondo [1,10], Thomas J. Hope [3], Francois Villinger[2] & Elena Martinelli [1] ✉

HIV-1 persistence during ART is due to the establishment of long-lived viral reservoirs in resting immune cells. Using an NHP model of barcoded SIV-mac239 intravenous infection and therapeutic dosing of anti-TGFBR1 inhibitor galunisertib (LY2157299), we confirm the latency reversal properties of in vivo TGF-β blockade, decrease viral reservoirs and stimulate immune responses. Treatment of eight female, SIV-infected macaques on ART with four 2-weeks cycles of galunisertib leads to viral reactivation as indicated by plasma viral load and immunoPET/CT with a $^{64}Cu$-DOTA-F(ab')$_2$-p7D3-probe. Post-galunisertib, lymph nodes, gut and PBMC exhibit lower cell-associated (CA-)SIV DNA and lower intact pro-virus (PBMC). Galunisertib does not lead to systemic increase in inflammatory cytokines. High-dimensional cytometry, bulk, and single-cell (sc)RNAseq reveal a galunisertib-driven shift toward an effector phenotype in T and NK cells characterized by a progressive downregulation in TCF1. In summary, we demonstrate that galunisertib, a clinical stage TGF-β inhibitor, reverses SIV latency and decreases SIV reservoirs by driving T cells toward an effector phenotype, enhancing immune responses in vivo in absence of toxicity.

Interruption of antiretroviral therapy (ART) leads to rapid rebound of viremia in the vast majority of people living with HIV-1 (PLWH) due to the establishment of a persistent HIV-1 reservoir early after infection[1,2]. A key mechanism of this persistence is the ability of HIV-1 to enter a state of virological latency characterized by the silencing of viral gene expression and/or lack of viral proteins translation[3,4]. This allows the virus to remain invisible to the immune system and latently infected cells to survive and proliferate by homeostatic or antigen-driven

[1]Department of Medicine, Division of Infectious Diseases, Feinberg School of Medicine, Northwestern University, Chicago, IL, USA. [2]New Iberia Research Center, University of Louisiana at Lafayette, New Iberia, LA, USA. [3]Cell and Developmental Biology, Feinberg School of Medicine, Northwestern University, Chicago, IL, USA. [4]AIDS and Cancer Virus Program, Frederick National Laboratory for Cancer Research, Frederick, MD, USA. [5]Division of Immunology, Tulane National Primate Research Center, Covington, LA, USA. [6]Center for Synthetic Biology, Northwestern University, Chicago, IL, USA. [7]Robert H. Lurie Comprehensive Cancer Center, Feinberg School of Medicine, Northwestern University, Chicago, IL, USA. [8]Laboratory of Immunoregulation, National Institute of Allergy and Infectious Diseases, National Institutes of Health, Bethesda, MD, USA. [9]Department of Microbiology and Immunology, Tulane University School of Medicine, New Orleans, LA, USA. [10]Center for Pathogen Genomics and Microbial Evolution, Northwestern University Havey Institute for Global Health, Chicago, IL, USA. [11]These authors contributed equally: Deepanwita Bose, Mariluz Araínga. ✉e-mail: elena.martinelli@northwestern.edu

proliferation[5,6]. Of note, the viral reservoir was initially thought to be stable. However, recent evidence suggests that stochastic HIV reactivation under ART occurs, and selective killing is favored in cells bearing replication competent virus integrated in transcriptionally active sites within the genome[7–9]. Hence, integration site, but especially the activation status of the infected cells profoundly influences HIV-1 persistence.

Ongoing efforts to achieve a functional cure for HIV-1 are directed towards supplementing ART with immunotherapies targeting the viral reservoir. Such strategies, under the umbrella of "shock and kill", aim to reactivate the replication competent reservoir and eliminate latently infected cells by viral cytopathic effects or immune-mediated killing[10,11]. However, these strategies have thus far failed to achieve a reduction of the viral reservoir or post-ART virologic control in either the clinic or preclinical models. This is mainly due to the low inducibility of latent proviruses and the heterogeneity of the mechanisms of persistence[12–15]. In contrast, therapeutic vaccination strategies focused on enhancing HIV/SIV-specific responses have had some discreet measure of success[16]. However, currently, no single strategy leads simultaneously to latency reversal and stimulation of effective immune responses.

The HIV life cycle and HIV's ability to replicate efficiently are especially dependent on the activation status of the infected cells. In this context, significant advances have been made to directly activate immune cells through non-canonical pathways in order to promote HIV latency reversal[17]. However, the activation and differentiation of immune cells is intrinsically linked to cellular metabolism[18]. Indeed, signaling pathways that govern immune cell differentiation and activation such as mTOR and Wnt/β-catenin, not only have been implicated in regulating HIV latency[19–21], but are also critical regulators of cell metabolism[22]. The metabolic status of an infected cell, in turn, plays a critical role in its ability to support latency reactivation and viral production[19,23].

In this context, recent evidence suggests that TGF-β plays a critical role in the regulation of immune cell activation and metabolic reprogramming[24–26]. Specifically, in the context of CD8[+] T cells, TGF-β has been shown to suppress mTOR signaling preserving the metabolic fitness of memory CD8[+] T cells[25] and stem-like antigen specific CD8[+] T cells through its modulation of Wnt/β-catenin factor TCF1[26,27]. TCF1 and the mTOR pathway are also critical to T cell differentiation and responsible for the transition from activated effector cells to resting memory cells during LCMV infection[22,26]. The regulation of quiescence in CD8[+] T cells that follows continuous TGF-β stimulation is critical to the transition to a memory phenotype and it is driven by specific metabolic changes that are linked to decreased glycolytic activity, more efficient mitochondrial respiration, and long-term survival[25,26].

Similarly, in the context of NK cells, TGF-β has been implicated in decreasing their baseline metabolism driving lower expression of markers of NK cytotoxic activity[28].

While TGF-β-mediated suppression of TCR and IL-2 signaling were shown to lead to lower CD4[+] T cell activation following cognate antigen recognition in older studies[24,29], more recent work using CD4[+] T cell-specific deletion of the TGF-β receptor demonstrated an even more profound effect of TGF-β on all stages of CD4[+] T cell activation, proliferation and cytotoxic response to LCMV than in CD8[+] T cells[30]. However, little is known on the specific role of TGF-β in regulating the transition to and from memory and effector phenotypes in CD4[+] T cells and how this may be associated with TGF-β-driven changes in CD4[+] T cell metabolism. Moreover, TGF-β regulates the expression of CD103 and other surface and intracellular factors essential to T cell residency in mucosal tissues[31–33]. Hence, TGF-β is considered the master regulator of mucosal immunity[34].

We and others have recently demonstrated that TGF-β regulates HIV-1 latency in primary CD4[+] T cells ex vivo and in vivo[35–37]. Latency reversal was detected in a non-human primate (NHP) model of HIV infection following a short treatment with a clinical stage TGF-β inhibitor, galunisertib (LY2157299)[38]. In that study, we documented latency reversal particularly at the level of the gut mucosal tissue using the [64]Cu-anti-gp120 Fab$_2$(7D3) probe and immuno-PET/CT[35]. We further validated the ability of immune-PET/CT to identify sites of viral reactivation and replication in the gut by performing tissue resection in hot areas of the gut identified by PET followed by confirmatory PCR for vDNA/RNA and vRNAscope[35].

Herein, we demonstrate how treatment with galunisertib with a 2-week on, 2-week off regimen that mimics the therapeutic regimen employed in the clinic in phase 1 and 2 trials of solid tumors[39,40], leads to profound transcriptional and functional changes in immune cells in the absence of overt toxicity or increased systemic inflammation. Importantly, we observed a shift toward a transitional effector phenotype in CD4[+] T cells and other immune cells both systemically and in the lymph nodes. This shift was accompanied by, and likely responsible for, increased viral reactivation in SIV-infected, ART-treated macaques documented by molecular techniques and PET/CT images. At the end of the treatment with galunisertib, we detected lower viral reservoir levels, including total and intact proviral DNA in both PBMC, gut and lymph nodes and significantly higher immune responses.

## Results

### 2-weeks on-off therapeutic regimen with galunisertib leads to viral reactivation in SIV-infected, ART-treated macaques

To confirm galunisertib-driven HIV/SIV latency reversal and investigate the underlying mechanisms, 8 Indian origin rhesus macaques (*Macaca mulatta*, Mamu-A01-, -B08, -B17-, all females) were infected intravenously with 300 TCID50 of the barcoded SIVmac239M2. We initiated ART treatment (daily co-formulated Tenofovir [PMPA], Emtricitabine [FTC] and Dolutegravir [DTG]) on week 6 post-infection (pi). A 2-week on, 2-week off therapeutic cycle with galunisertib (20 mg/Kg twice/ daily orally) started at week -35pi and continued for a total of 4 cycles (Fig. 1A and Supplementary Table S1). ART was discontinued 3 weeks after the last galunisertib dose, and the macaques were followed for 6 weeks after ART discontinuation. The median peak plasma viral load (pVL) was 10[8] copies/mL at week 2pi. Given the synergistic activity of anti-PD1 and anti-TGF-β therapies in cancer[41,42], a rhesus anti-PD1 antibody was administered at 5 mg/kg before the 3[rd] and 4[th] cycle to 2 macaques (08M156 and A6X003). However, no differences were noted for these 2 macaques in any of the parameters we measured, and the data were pooled.

Full suppression to undetectable levels (pVL LOD 15 copies/mL) was achieved in 3 out of the 8 macaques at week 10pi. In the other 5 macaques, pVL fell below 65 copies/mL by week 22pi with a single blip of 400 copies/mL in A8T010 at week 29pi (Fig. 1B). Following the start of the galunisertib treatment, pVLs increased in 7 out of 8 macaques from a single peak over undetectable in 08M171 and A8L057 to several peaks and up to 10[3] copies/mL in the other macaques. Of note pVLs in A8T010 and A8R095 were undetectable for over 5 weeks before, respectively, blips of up to 10[2] copies/mL were detected following galunisertib treatment initiation (Fig. 1B). More frequent blips were noted during the first 2 cycles with galunisertib compared to cycles 3 and 4 (Fig. 1C). However, 08M171 and A8L057 did not experience a pVL increase until the 4[th] cycle.

Importantly, in support of the pVL data above, we documented viral reactivation also using immunoPET/CT. The [64]Cu-anti-gp120 Fab$_2$(7D3) probe was injected 24hrs before each scan and scans performed before the first and after the last galunisertib dose in each cycle. As shown in Fig. 2, Supplementary Fig. S1 and Supplementary Movies S1–S8, the PET signal visibly increased in different tissue areas after cycle 2 (in A8R095, 08M156, A8L014 and A8T010) or at the beginning of cycle 3 (in A6X003, 08M134 and A8L057). In 08M171 we observed an increase in the gut area only at the beginning of cycle 2. An unforeseen issue with probe stability in cycle 3 led to exclusion of the

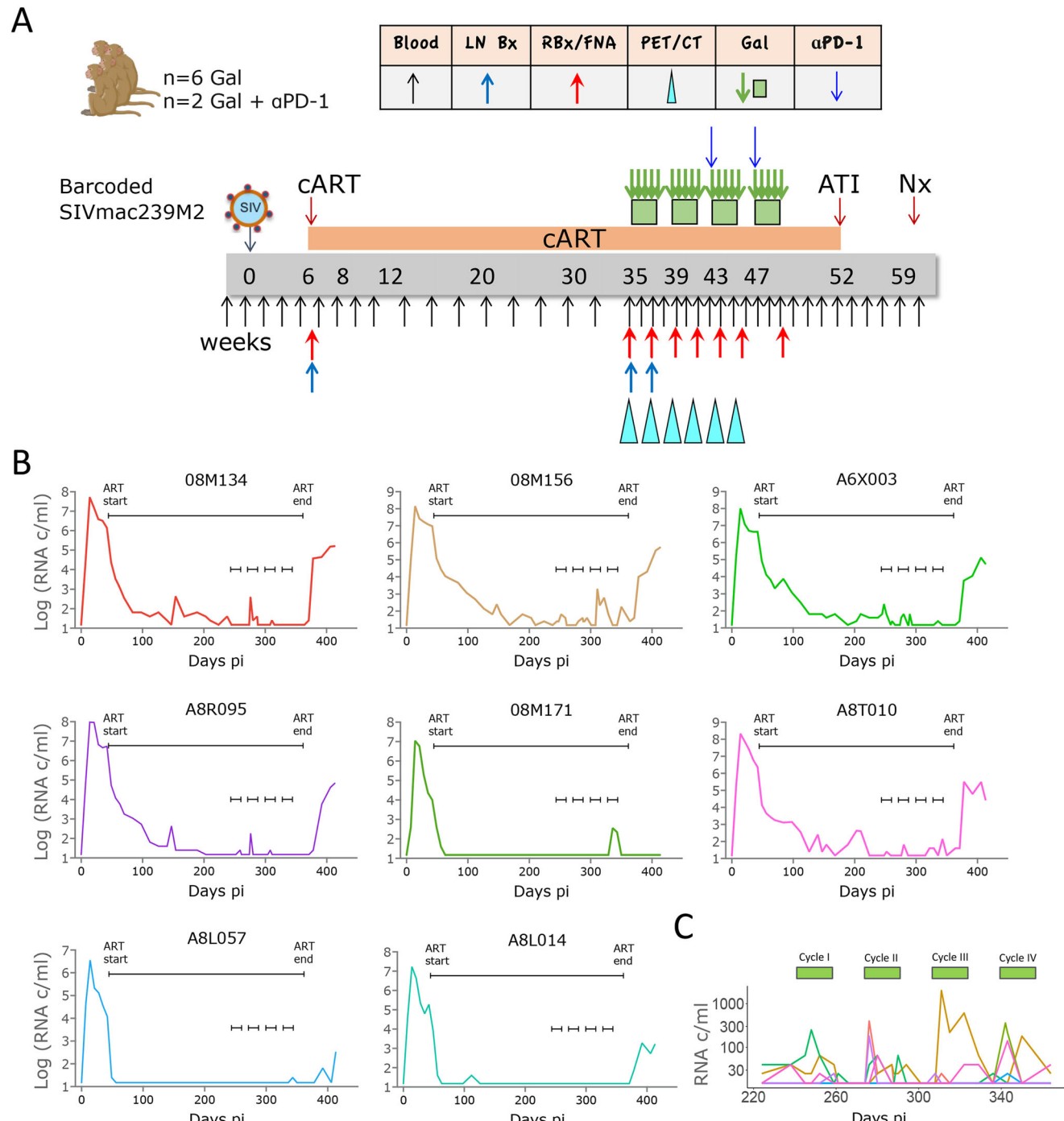

**Fig. 1 | Four 2-weeks cycles with galunisertib lead to viral reactivation in blood.** **A** Schematic representation of the study and sampling schedule. **B** Plasma VL in blood for each macaque throughout the study. The longer black line indicates the period on ART, while the 4 small black lines indicate the start and end of each galunisertib cycle. **C** Enlarged plasma VL for all macaques during galunisertib therapy. Green bars indicate galunisertib cycles. Source data are provided as a Source Data file. Image from BioRender.

last 2 scans of 08M171 from the analysis (Supplementary Fig. S1, 08M171 infection, treatment and scan were offset compared to the other macaques). A corresponding increase in mean standard uptake values (SUV) was detected in the gastrointestinal area and axillary lymph nodes (Fig. 2B) and was significant in cycle 3 compared to before cycle 1 (BC1). In these anatomical areas (ROIs in Supplementary Fig. S2 and Supplementary Movies 9-10), SUV increases likely correspond to increases in viral replication as demonstrated in previous studies[43,44]. A PET signal increase was also noted in the area of the vertebral column (spine) and nasal associated lymphoid tissues

(NALT). However, neither cerebrospinal fluid (CSF) nor bone marrow (BM) or NALT tissue were collected during the study, and we have no prior validation of the specificity of the signal in these anatomical locations. Hence, whether this signal corresponds to increased viral replication and whether this occurs in the vertebral bones or cerebrospinal fluids remain to be determined. No SUV increase was present at the level of the spleen or kidney, where probe accumulation and background signal likely masked any specific signal. However, a significant increase in SUV was detected in the liver (Fig. 2B). Similar increases in PET signal are also evident when considering the SUV Total

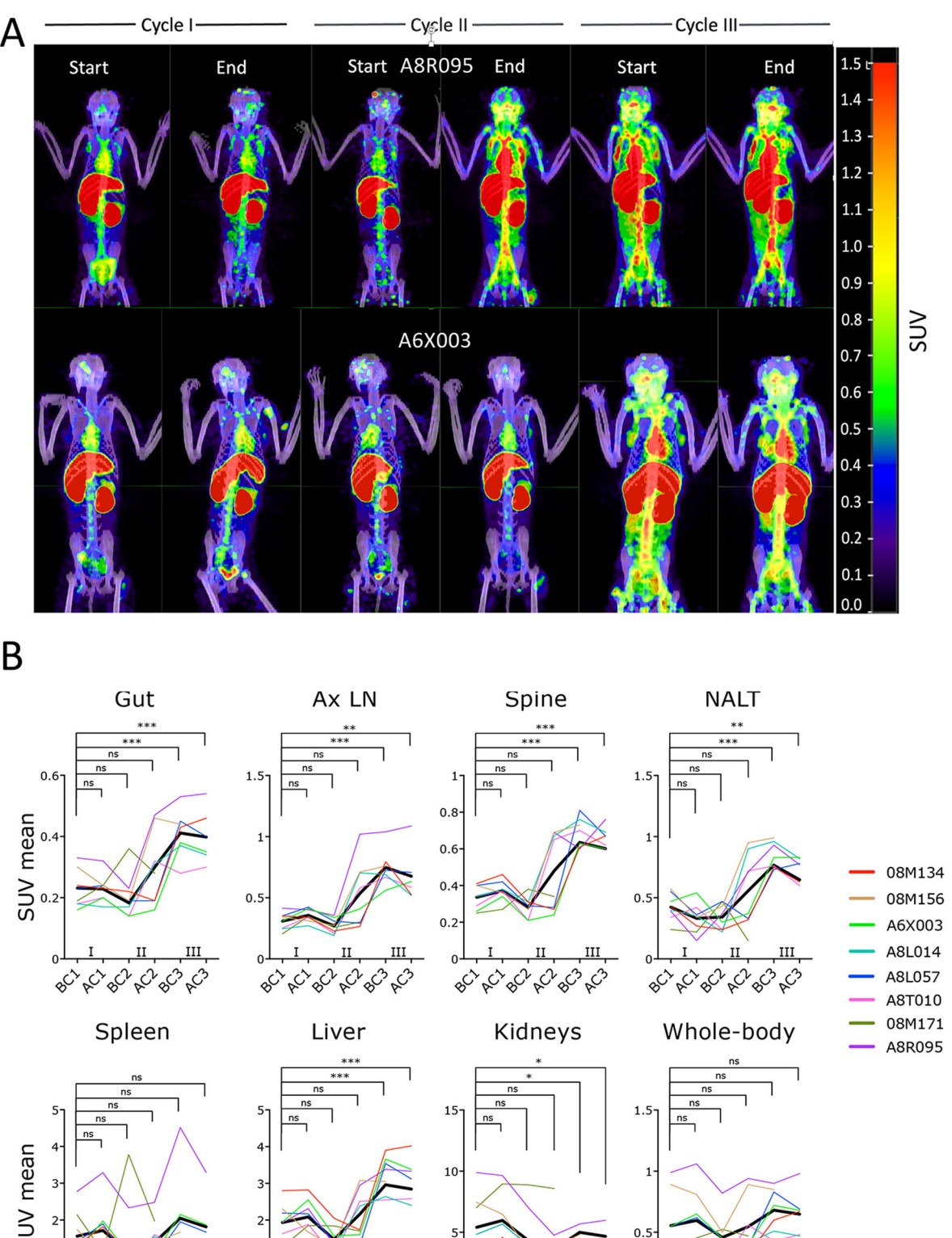

**Fig. 2 | Galunisertib leads to viral reactivation in tissues. A** The $^{64}$Cu-DOTA-Fab$_2$(7D3) probe was injected ~24 h before PET/CT scan before and at the end of each of the first 3 galunisertib cycles. Representative images from the maximum intensity projections (MIP) of fused PET and CT scans are shown for a macaque with a major increase at the end of cycle 2 and one showing increase at the beginning of cycle 3. MIPs were generated using the MIM software, set to a numerical scale of 0–1.5 SUVbw, and visualized with the *Rainbow* color scale. **B** Mean SUV were calculated for each anatomical area, and values were analyzed with mixed-effect analysis. Data from the scans performed at the last 2 time points (BC3 and AC3) in 08M171 were excluded because of technical issues with the probe. Thicker black line represents the mean. *P*-values were calculated for comparison of each time point with the before cycle 1 time point (BC1; AC1= after cycle 1, BC2= before cycle 2; AC2= after cycle 2; BC3= before cycle 3; AC3= after cycle 3; Holm-Sidak multiple comparison correction; *$p \leq 0.05$ **$p \leq 0.01$ ***$p \leq 0.01$). Source data are provided as a Source Data file.

in these anatomical areas (Supplementary Fig. S3A). Moreover, blood pool activity (BPA) also increased during the 3rd cycle (Supplementary Fig. S3B). Whether this was due to galunisertib-specific effects on probe pharmacokinetics, changes in viral antigen or probe-antigen kinetics remains to be determined. However, when the mean SUV was normalized for BPA in the gut, axillary lymph nodes and spine, the signal increase in the 3rd cycle was lost, but an increase during the first cycle became evident (Supplementary Fig. S3C). Of note, the increase in non-BPA normalized SUVmean in the gut and lymph node areas in most cases followed an increase in cell-associated vRNA detected, respectively, in colorectal biopsies and fine needle aspirates (FNA) at the same time points during treatment (Supplementary Fig. S4).

### Decreased viral reservoir in absence of systemic inflammation after 4 cycles with galunisertib

To determine the impact of galunsiertib on SIV reservoir, we measured CA-vDNA in PBMC, colorectal biopsies and lymph nodes (LN). A significant decrease in CA-vDNA was detected in all tissues between week 35pi (beginning of cycle 1, BC1) and week 49pi (after/end of cycle 4, AC4) for gut and LN, and between week 35pi (BC1) and end of cycle 3 (AC3) for PBMC (Fig. 3A, AC4 not measured for PBMC and Supplementary Fig. S5A). In the gut and LN (right axillary), decreases ranged from a Log to 1/3 of a Log (gut: median 0.77; range: 0.33–0.94 fold LN: median: 0.93; range: 0.33–0.98 fold decrease). In the PBMC the decrease was slightly less pronounced with a median half Log decrease (median: 0.58; range 0.28–0.88 fold decrease). However, the comparison for the PBMCs was between week ~35 and week ~45 (end of cycle 3, AC3) instead of the end of all 4 cycles, because a snap frozen pellet was not available at the end of cycle 4 for PBMCs. The SIV reservoir in PBMCs was also monitored by SIV-IPDA (intact proviral DNA assay) comparing before cycle 1 (BC1) to the end of cycle 4 (AC4). Of note, we observed significant decreases of both total and intact provirus by SIV-IPDA. Intact provirus declined similarly to the CA-vDNA with a median of half Log (median: 0.53; range: 0–0.71 fold decrease).

In contrast, no decline in CA-vDNA was detected in the PBMCs of a group of 4 macaques infected intravenously with the same stock of SIVmac239M2 for a separate study. These 4 macaques were placed on ART on week 6pi as in our study. However, they were infected several months after our study and samples collected at similar time points varied in their availability. No decline in CA-vDNA was detected under ART, between weeks 28 and 52pi (untreated group, Supplementary Fig. S5B). This suggests that the decline in CA-DNA in our study was not due to ART alone. However, in absence of an appropriate control group, it is not possible to be determine with confidence the relative contribution of ART and galunisertib to the decline.

Importantly, we found no significant changes in any of the clinical variables (chemistry and hematology, Supplementary Data 1 and 2) measured before, during and after the galunisertib therapy. Moreover, we observed no changes in the concentrations of inflammatory chemokines and cytokines measured in plasma before and after the first 2 treatment cycles and at the end of the last cycle (Fig. 3C). The only difference in cytokine and chemokine levels after galunisertib treatment was a small increase in IL-10 detected at the end of the last cycle compared to before treatment (Supplementary Fig. S5C).

### Galunisertib treatment drives an effector phenotype in T and NK cells

The phenotype of PBMCs before and after Galunisertib treatment was monitored by high-parameter flow cytometry of T and NK cell subsets and phenotype (Supplementary Table S2). Classical subsets and single-color analysis of MFI revealed a substantial increase in the expression of CD95 and a profound consistent decrease in TCF1 expression in CD4+ T cells that continued throughout the treatment (Fig. 4A). In contrast, a small decrease in CD62L after the first cycle,

reverted to baseline during the following cycles. The frequency of naïve cells, defined as CD95- within CD4+ T cells, decreased in parallel with the increase in CD95 (gating strategy in Supplementary Fig. S6). Interestingly, there was no change in the expression of CCR7 or CD28 within CD95+ CD4+ T cells (Fig. 4B). Hence, the frequency of central memory and effector memory as defined by CD95 and CD28 or CCR7 did not change (Supplementary Data 3). The levels of T-bet did not change significantly (Supplementary Fig. S7A). However, we detected a downregulation of the gut homing receptor integrin α4β7 and a decrease in the levels of granzyme B (GRZB, Supplementary Fig. S7A). Of note, the expression of activation markers CD69, HLA-DR and Ki67 remained mostly unchanged, with the exception of an increase in HLA-DR at the beginning of cycle 2 compared to before galunisertib (Fig. 4C). Interestingly, the effect of Galunisertib on CD8+ T cells was not as pronounced as it was on CD4+ T cells. No significant increase was detected in CD95 expression and TCF1 downregulation reached significance only at the end of the treatment (Fig. 4D). Markers of cell activation like HLA-DR, Ki67 and CD69 did not change (Fig. 4D and Supplementary Fig. S7B). However, we detected a significant decrease in T-Bet at the beginning of cycle 2 (BC2) compared to before treatment (BC1). Finally, in contrast to CD4+ T cells, the decrease in GRZB was more pronounced at several time points during treatment and there was a sustained significant increase in CCR7 expression in memory CD8+ T cells (Fig. 4E). Interestingly, the expression of PD1 on both CD4+ T cells and CD8+ T cells either did not change or was slightly downregulated. Importantly, the frequency of PD1+ CD101+ (TCF1low) exhausted memory CD8+ T cells[45] did not change (Supplementary Fig. S7B). Finally, within NKG2A+ CD8+ NK cells, there was a pronounced increase in CD16 expression and an initial increase in the proliferation marker Ki67 (AC1 vs BC1; Fig. 4F).

High-dimensional data visualization with tSNE and clustering analysis with FlowSOM[46] confirmed the results of the classical analysis. We performed tSNE and FlowSOM after data clean up with FlowClean and normalization with the SwiftReg algorithm[47]. We performed two analyses. One analysis compared before (BC1) and after the last cycle (AC4) only (Fig. 4G and Supplementary Fig. S8A). A second analysis was performed on all time points (Supplementary Fig. S8B, C). When we compared only BC1 and AC4, the PhenoGraph clustering algorithm identified 36 populations. FlowSOM with 36 populations identified 6 populations of CD4+ T cells, 6 of CD8+ T cells and 4 of NK cells (NKG2Ahigh CD8+; Supplementary Fig. S8A). The remaining populations were likely monocytes and other minor subsets. Direct comparison of each population revealed a decrease in Pop 31 (naïve CD8 T cells) and Pop 30 (central memory Ki67+ CD8+ T cells), and an increase in Pop 2 and 3 (effector and central memory CD4+ T cells). Finally, there was a decrease in HLA-DR high Pop0 and an increase in CD16high NKG2A- Pop14 (Fig. 4H and Supplementary Fig. S8A). Visual inspection of the tSNE plots (Fig. 4G) revealed 3 areas mostly occupied by cells in the post-galunisertib AC4 group (New1, 2 and 3) which were characterized by high levels of CD16 and GRZB (New 3 is likely NK cells). In contrast, 3 areas mostly occupied by cells in the pre-galunisertib group BC1 (Old1, 2 and 3) were characterized by high levels of TCF1 and low CD95, confirming the finding of the classical analysis. Analysis of all time points with phenograph-derived 38 populations in FlowSOM recapitulated findings obtained with the classical analysis and the BC1 AC4 comparison with no additional insights (Fig. 4I).

### Galunisertib treatment in vivo increases pTreg while decreasing pTfh

The frequencies of circulating Tregs and Tfh cells were monitored with an established flow cytometry panel[48,49] after the first and third cycles of galunisertib. The frequency of all CD4+ Tregs (CD25high FoxP3+; gating in Supplementary Fig. S9) and CD8+ Tregs increased

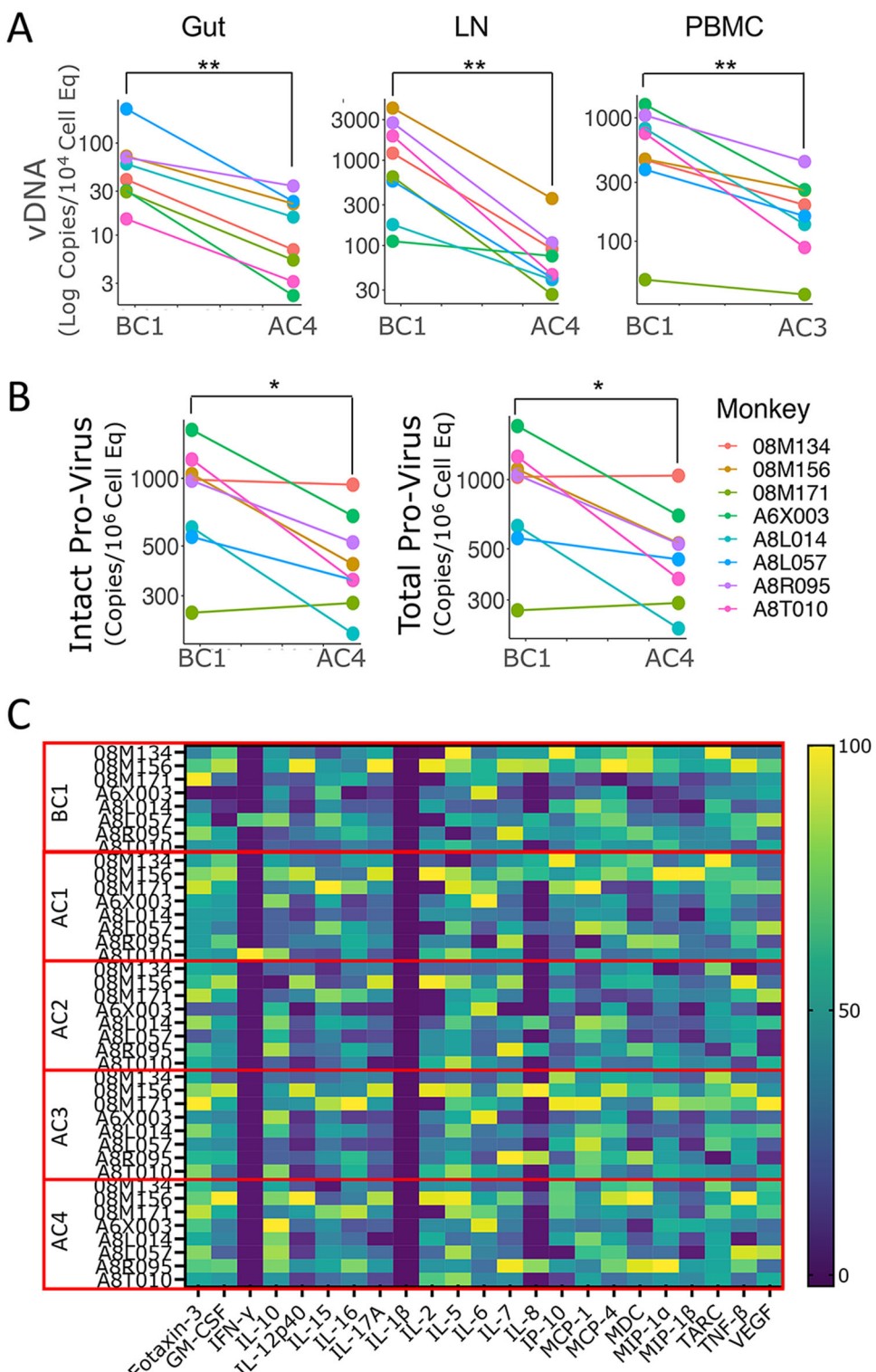

**Fig. 3 | Galunisertib decreases viral reservoir in absence of systemic inflammation. A** Levels of cell-associated (CA)-vDNA per cell equivalent are shown for the time point before cycle 1 (BC1) and at the end of cycle 3 (AC3) or 4 (AC4) for the respective tissues for all 8 macaques. **B** IPDA data are shown for intact and total provirus for BC1 and AC4 in PBMC for the 8 macaques. *P*-values are shown for Wilcoxon matched pair signed-rank non-parametric two-tailed test comparing before and after galunisertib data from the 8 macaques (*$p \le 0.05$ **$p \le 0.01$ ***$p \le 0.01$). **C** Heat map of cytokine concentration in plasma at the indicated time point are shown after Log transformation and normalization. Statistical analysis was run on each factor separately and together (no significant differences after multiple comparison adjustment). Source data are provided as a Source Data file.

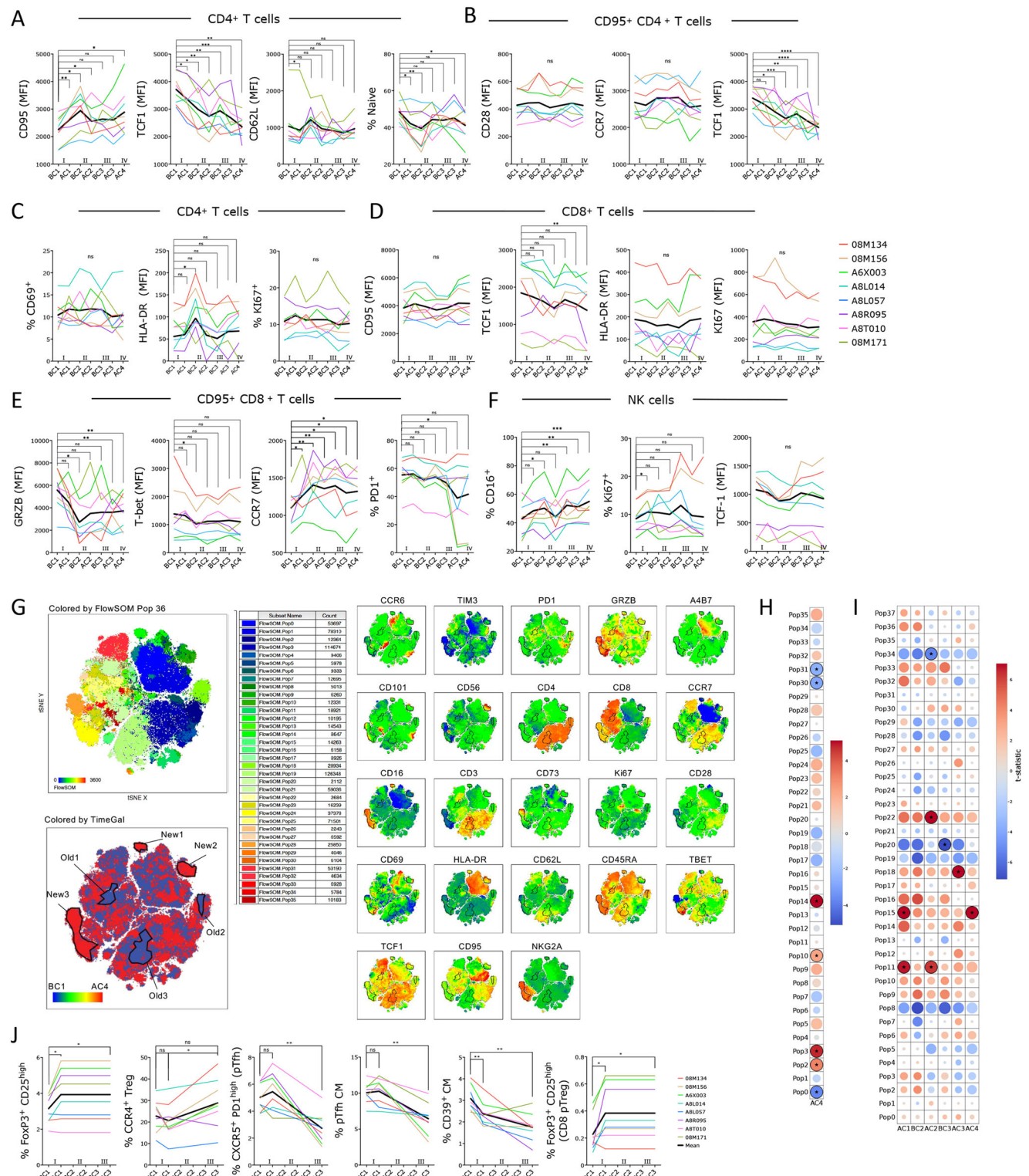

after the first cycle and remained higher until the end of the 3rd cycle, while CCR4+ Treg were proportionally higher at the end of the treatment compared to before (Fig. 4J). In contrast, circulating Tfh (CXCR5+ PD1+; gating in Supplementary Fig. S9) were lower at the end of the Galunisertib treatment, both within total and central memory CD4+ T cells (Fig. 4J). Finally, the expression of CD39 (ecto-nucleotide triphosphate diphosphohydrolase 1), which tracks within extracellular adenosine and immunosuppressive effects, was lower on total and central memory CD4+ T cells (Supplementary Data 3 and Fig. 4J, respectively).

## Bulk RNAseq of PBMC and single-cell (sc)RNAseq analysis of LN confirm a profound shift toward an effector phenotype

In our previous studies, we determined that 6hrs after galunisertib treatment in naïve macaques, there was an upregulation of the AP1 complex (JUN and FOS) and several genes encoding ribosomal proteins in CD4+ T cells[44]. To understand the early and later effects of galunisertib in the context of SIV infection, in the current study, we performed bulk RNAseq of PBMC isolated 1 h after the first administration of galunisertib in cycle 1 and at the end of the 1st 2-weeks cycle. We found 640 genes significantly modulated (FDR < 0.05; abs(log2

**Fig. 4 | Galunisertib leads toward effector in T and NK cells, increasing Treg and decreasing Tfh frequencies. A–F** Geometric mean fluorescent intensities (MFI) of each marker and frequency of indicated subset within live, singlets CD3$^+$ CD4$^+$ T cells (**A** and **C**) or CD3$^+$ CD4$^+$ CD95$^+$ T cells (**B**) or CD8$^+$ or CD8$^+$ CD95$^+$ T cells or NK cells (NKG2A$^+$ CD8$^+$ CD3$^-$ cells) are shown. Thick black line represents the mean. Changes from baseline (beginning of cycle 1, BC1) are shown for graphs with at least 1 significant difference (Repeated measures ANOVA with Holm-Sidak correction for multiple comparisons; *$p \leq 0.05$ **$p \leq 0.01$ ***$p \leq 0.01$). **G** tSNE of lymphocyte, live, singlets events after normalization for BC1 and AC4 (all 8 macaques) with FlowSOM 36 clusters overlaid on tSNE (top left) or heatmap of each markers MFI (right) or heatmap of time point (blue is BC1 and red is AC4; bottom left) is shown. 6 populations were manually gated on red or blue areas (red, New1-3 and blue Old1-

3). **H** Bubble chart displaying changes in AC4 from BC1 in populations (FlowSOM clusters) characterized by markers MFI in Supplementary Fig. S8A. Color is proportional to the effect size and size to *p*-value (Wilcoxon sum rank non-parameter two-tailed test). **I** Bubble chart displaying changes from BC1 at all time points in populations (FlowSOM clusters) characterized in Supplementary Fig. S8C (ANOVA repeated measures with Holm-Sidak multiple comparisons correction; *$p \leq 0.05$ **$p \leq 0.01$ ***$p \leq 0.01$). **J** Frequency of indicated subset within live, singlets CD3$^+$ CD4$^+$ T or CD3$^+$ CD4$^+$ CD95$^+$ CD28$^+$ T cells (CM = central memory) or within CD3$^+$ CD8$^+$ T cells. Changes from baseline (BC1) are shown (ANOVA repeated measures with Holm-Sidak correction for multiple comparisons; *$p \leq 0.05$ **$p \leq 0.01$ ***$p \leq 0.01$). Source data are provided as a Source Data file.

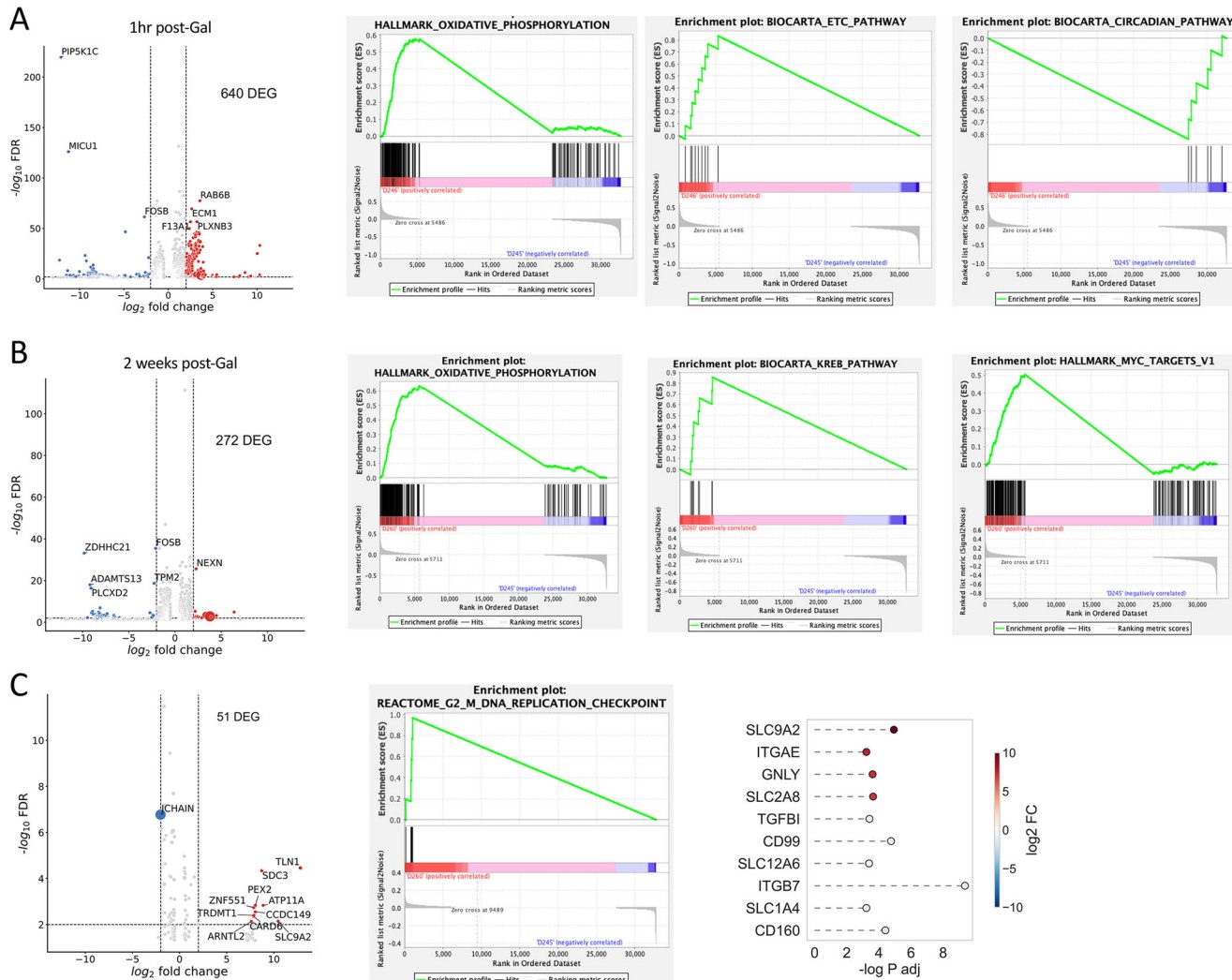

**Fig. 5 | OXPHOS and other metabolic pathways increase rapidly with TGF-β blockade.** Bulk RNAseq was performed with PBMC from before cycle 1 (24 h) and 1hrs after the first dose of galunisertib (**A**) or after the last dose of cycle 1 (**B**) and with rectal biopsies collected before cycle 1 (24 h) and after the last dose of cycle 1 (**C**). The number of differentially expressed genes (DEG) obtained by DESeq2 with

an FDR < 0.05 and abs(log$_2$FC)>2 are shown in each respective volcano plot. Enrichment plots are shown after GSEA (with all FDR < 0.05 DEGs) for significantly enriched pathways (top 1 or 2 pathway by ES). **C** Lollipop graph of selected DEG of interest among significantly different genes (FDR < 0.05). Source data are provided as a Source Data file.

FC(Fold Change))>2) in PBMC just 1 hr after the first dose of galunisertib. The majority (457 genes) were downregulated (Fig. 5A). Gene set enrichment analysis (GSEA) revealed an upregulation of the oxidative phosphorylation (OXPHOS; Enrichment Scores (ES) 0.58 FDR = 0.048) and the reactive oxygen (ES 0.55 FDR = 0.075) pathways among the Hallmark genet sets (Fig. 5A and Supplementary Fig. S10). Among the Biocarta sets, there was an enrichment in the electron

transport chain (ETC; ES 0.84 FDR = 0.132) and a downregulation of the circadian (ES −0.81 FDR = 0.163) pathways (Fig. 5A and Supplementary Fig. S10). Finally, among the top modulated genes (by FC), we identified several genes encoding for soluble transporters, while classical activation markers like CD69 and CD38 were downregulated (Supplementary Fig. S10B). An early engagement of metabolic pathways was confirmed by enrichment analysis of significant DEG with

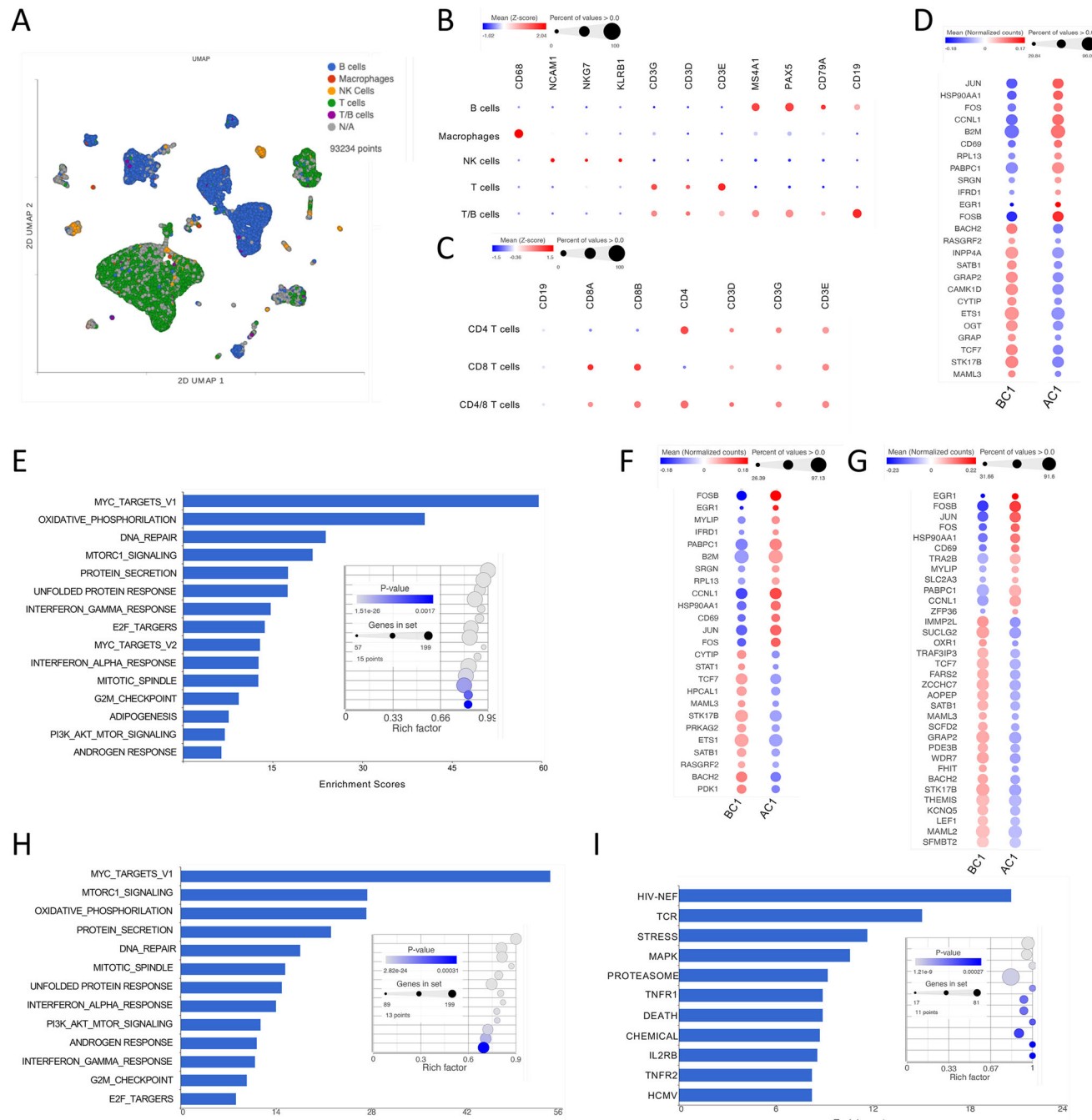

**Fig. 6 | scRNAseq of lymph node before and after cycle 1 confirms a switch toward effector and increased metabolism in all immune subsets with galunisertib. A** UMAP projection of 93234 cells from lymph nodes collected right before and at the end of cycle 1 from all 8 macaques (16 samples). Gene-based classification of major immune subset is overlaid on UMAP. In gray are unclassified cells. Bubble plots showing expression (mean normalized counts proportional to the color; size proportional to the percentage of cells) of each marker listed in each cell subset. Marker listed are those used for classification of major immune subsets

(**B**) or CD4[+] and CD8[+] T cells (**C**). **D** Significantly different genes obtained by Hurdle model (FDR < 0.05; log2FC = 0.15) in the T cell subset are shown with color proportional to normalized counts. **E** Significantly enriched pathways (FDR < 0.01) in T cells DEGs within the hallmark collection. Significantly different genes (FDR < 0.05; log2FC = 0.15) in the CD4[+] (**F**) and CD8[+] (**G**) T cell subset. Significantly enriched pathways (FDR < 0.01) in CD4[+] T cells DEGs within the hallmark (**H**) and biocarta (**I**) collections. Source data are provided as a Source Data file.

Metascape[50] with an upregulation of adipogenesis, OXPHOS and fatty acid metabolism (Supplementary Fig. S10C). Of note, 2 weeks after the beginning of galunisertib, metabolic pathways were still among the most enriched upregulated pathways in PBMCs (Fig. 5B and Supplementary Fig. S11A). Among the top upregulated genes there were CD44, CCR5, several integrins and GRZA and GRZB (Supplementary Fig. S11B). Finally, we performed bulk RNAseq of rectal biopsy tissue before and after the first cycle with galunisertib (Fig. 5C). There were

only 51 differentially expressed genes (DEGs; FDR < 0.05; $\log_2$FC = 2). GSEA analysis of all DEGs (FDR < 0.05) revealed the G2_M_DNA replication pathway highly enriched (ES 0.97 FDR = 0.012) within the Hallmark set. Among the most interesting changes, we observed a pronounced downregulation of integrin β7, in contrast to an increase in integrin αE (CD103; Fig. 6C), suggesting that TGF-β-driven increase in αE[51] may be driven by non-canonical TGF-β pathway signaling not blocked by galunisertib.

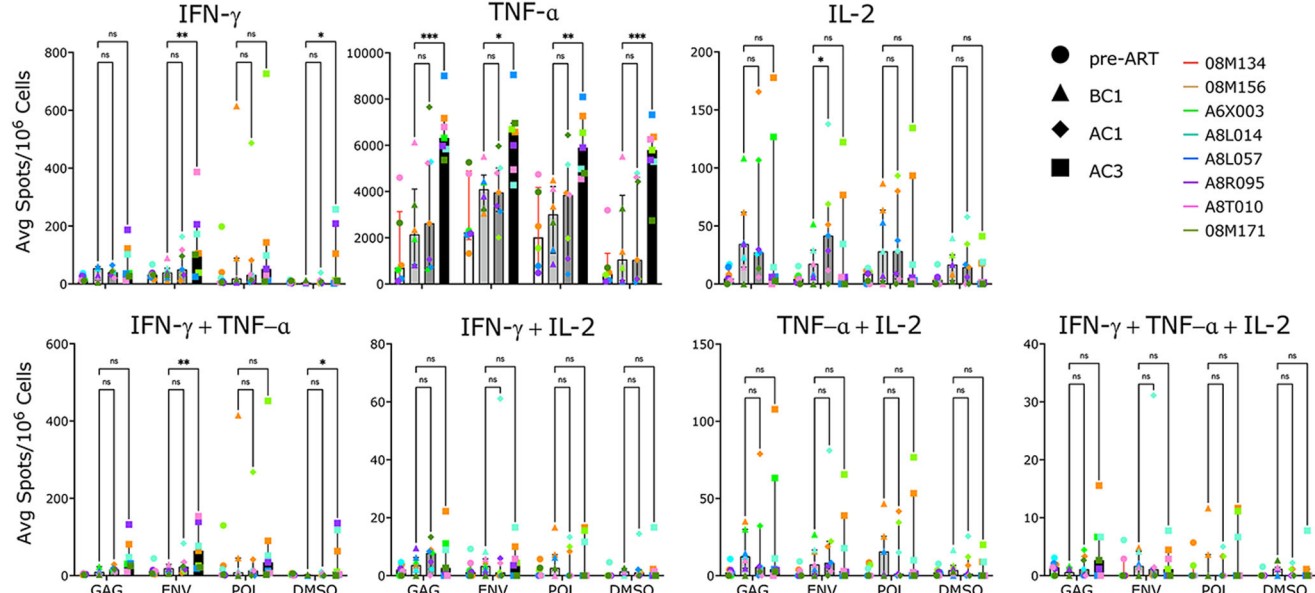

**Fig. 7 | Galunisertib increases SIV-specific responses.** Average spots (from triplicates) per $10^6$ PBMC at the time of ART initiation (pre-ART), before cycle 1 (BC1), after cycle 1 (AC1) and at the end of cycle 3 (AC3) with galunisertib after 24hrs ex vivo stimulation with 15-mer peptides (gag, env, pol) or mock (DMSO). Each post-galunisertib time point was compared to BC1 (Mixed effect analysis adjusted for multiple comparisons with Dunnet post-hoc p-values are shown; $*p \leq 0.05$ $**p \leq 0.01$ $***p \leq 0.001$). Bars represent the median with interquartile range as error bars. Source data are provided as a Source Data file.

To clarify the impact of galunisertib at the single cell level and in lymphoid tissues, we also performed scRNAseq analysis of cells isolated from LNs before (right axillar) and after (right or left inguinal) the first cycle (Fig. 6). Dimensionality reduction and clustering analysis with PCA and uniform manifold approximation and projection (UMAP)[52] was performed to visualize the data (Fig. 6A). However, cells were classified based only on gene expression (Fig. 6B, C). We first classified major subsets: T cells (38,705 cells), B cells (31,385 cells), NK cells (1,628 cells), macrophages (285) and cells expressing both CD19 and CD3 (942T/B cells) (Fig. 6A shows this classification over UMAP). Then we classified only CD4$^+$ and CD8$^+$ T cell subsets (Fig. 6C and Supplementary Fig. S12A) and B/T subsets including naïve and germinal center (GC) B cells and Tfh cells (Supplementary Fig. S12B, C). Cell number for all these subsets did not change with treatment (Supplementary Fig. S13). However, differential gene expression analysis of T cells revealed an upregulation of members of the AP1 complex, CD69, β2 macroglobulin and RPL13 among the most upregulated genes (Fig. 6D). Moreover, it confirmed a downregulation of TCF1 at the transcriptional level (*TCF7* gene; Fig. 6D). Gene enrichment analysis revealed again Myc_targets_V1, OXPHOS and mTORC1 as the most enriched hallmark pathways (Fig. 6E). Of note, among gene ontology cellular processes, RNA processing was the most highly enriched pathway, followed by intracellular transport and catabolic processes following right after (Supplementary Fig. S14A) confirming an increase in translation and metabolism within these cells.

Since we noticed substantial differences in the impact of galunisertib on CD4$^+$ T cells compared to CD8$^+$ T cells by flow, we focused the analysis on these subsets. In CD4$^+$ T cells we found only 25 DEGs with a log2FC = 0.15, while 34 DEGs were in CD8$^+$ T cells with more downregulated genes in the CD8$^+$ T cells compared to the CD4$^+$ T cells. The AP1 complex and *TCF7* were again upregulated and downregulated respectively in both CD4$^+$ (Fig. 6F) and CD8$^+$ T cells (Fig. 6G). However, STAT1 was more strongly downregulated in CD4$^+$ T cells. Enrichment analysis showed once again upregulation of Myc_targets_V1, OXPHOS and mTORC1 pathways in both CD4$^+$ and CD8$^+$ T cells (Fig. 6H and Supplementary Fig. S14B). Interestingly, the most enriched set among in Biocarta was the HIV-Nef pathway (Fig. 6I) demonstrating the

relevance of these galunisertib-driven changes to HIV cell cycle and transcription (highly enriched KREB pathway as well). The 2$^{nd}$ most enriched Biocarta pathway was TCR signaling, linking galunisertib to cell activation. Next, we analyzed changes in gene expression in Tfh cells. In this subset we obtained a similar number of DEG as in other T cells and myc_targets_V1 was still the most enriched hallmark pathway (Supplementary Fig. S14C, D). In B cells, the AP1 complex was again prominently upregulated, together with CD83, CD69 and MAMU-DR. Of note, more genes were modulated in B cells (44 genes with a log2FC = 0.15 and 18 with logFC0.2) than in T cells with some differences in enriched pathways (Supplementary Fig. S14F). Similar genes were modulated in GC B cells and naïve B cells with several more DEGs in naïve B cells than in GC cells (Supplementary Fig. S15A, B). Finally, 144 genes were modulated in NK cells and 39 genes in macrophages (Supplementary Fig. S15C, D; log2FC = 0.15). GRZB was prominently upregulated, but CD44 downregulated. In macrophages genes were mostly downregulated including TCF7L2 and KLF4 suggesting an increase in inflammatory phenotype and decrease in M2 polarization[53] (Supplementary Fig. S15D).

## Galunisertib increases SIV-specific responses and changes barcode distribution

In order to understand how galunisertib affected immune cell function and SIV-specific responses, we stimulated PBMC with 15-mer SIV peptides from SIVmac239 gag, pol and env for 24hrs on antibody coated Elispot plates. Because of sample availability, we probed before and after the first cycle and after the 3$^{rd}$ galunisertib cycle only. Interestingly, by the end of the 3$^{rd}$ cycle there was a significant increase in IFN-γ secretion both SIV-specific (particularly against env) and non-specific (DMSO control). A notable increase in TNF-α release was similar in response to Gag and non-specifically (Fig. 7). In contrast, IL-2 release appeared to increase slightly after the first cycle (non-significant), but remained unchanged with a slight decrease by the end of the 3$^{rd}$ cycle (Fig. 7).

In order to understand if these changes in immune responses in combination with latency reversal and switch toward an effector phenotype may have impacted viral population dynamics, we analyzed

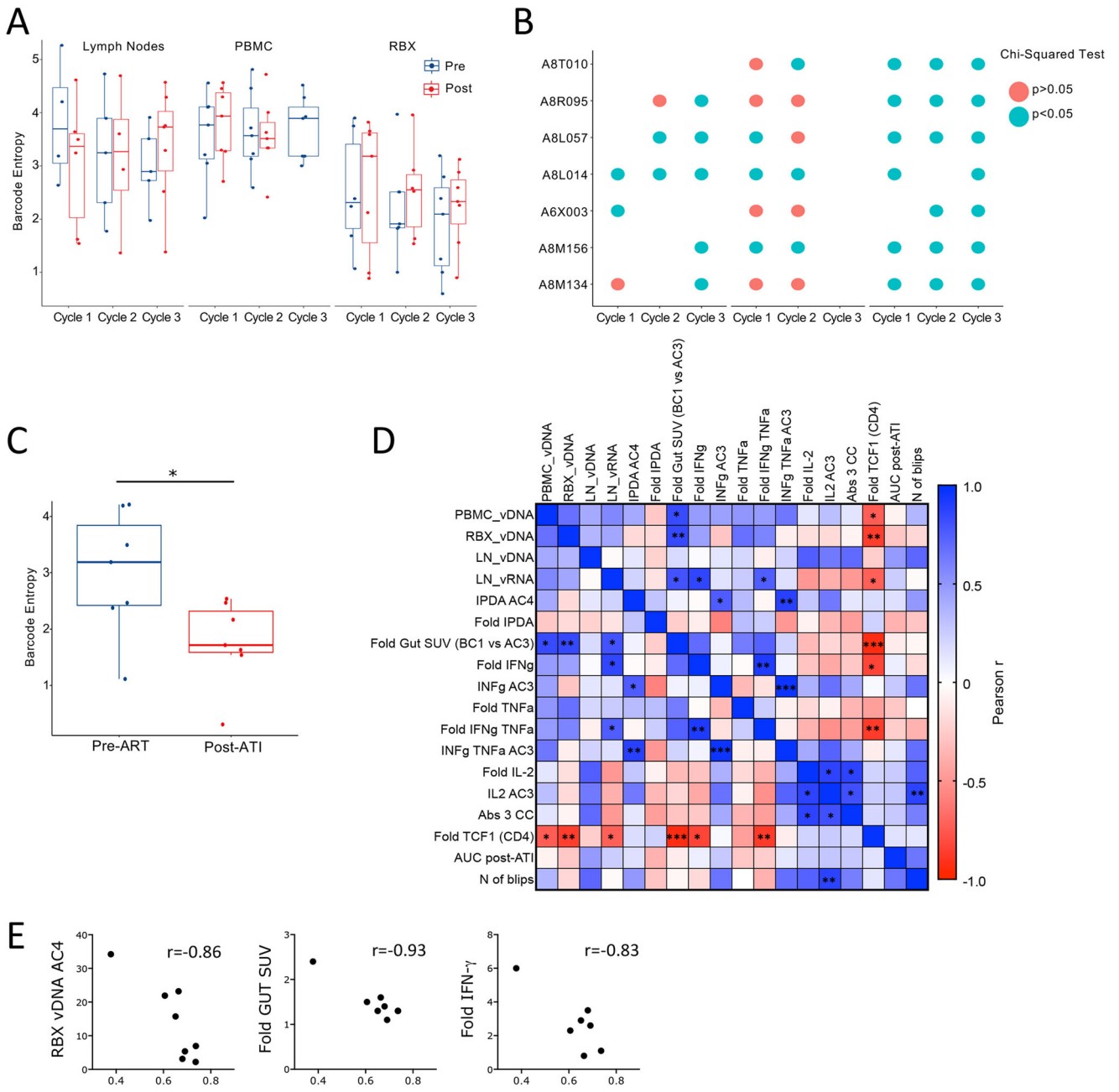

**Fig. 8 | TCF1 decrease associates with virological and immunological end-points. A** Barcode diversity measure as Shannon Entropy is shown before and after each of the first 3 galunisertib cycles for LN, PBMC and colorectal biopsies. Box-and-whisker plot represents the median +/− the interquartile range of data from 4 to 8 macaques (all data from macaques with detectable barcodes were included at each time point for a given tissue; no significant differences using linear mixed effects models). Blue= before; Red= end of each cycle. **B** Bubble plot shows the results of statistical testing (Chi-squared) for differences in frequency distribution of barcodes before compared to after, for each of the first 3 cycles of galunisertib

for each macaque in the indicated tissues. Blue indicates significant differences $p \leq 0.05$. **C** Barcode entropy of virus isolated at the time of ART initiation compared to week 6 post-ATI in plasma (Wilcoxon matched pairs two-tailed test; *$p \leq 0.05$). **D** Correlation matrix of several key variables of virological or immunological effect of galunisertib. Color is proportional to Pearson r coefficient. *$p \leq 0.05$ **$p \leq 0.01$ ***$p \leq 0.001$ indicate significant correlations. **E** Association between fold increase in TCF1 (MFI) from BC1 to AC4 with CA-vDNA levels at AC4, change in gut SUV at AC3 compared to BC1 and fold increase in IFN-γ (AC3 vs BC1). Person r is shown. All correlations have *$p \leq 0.05$. Source data are provided as a Source Data file.

changes in numbers and distribution of the viral barcodes before and after the first 3 treatment cycles. There was no barcode amplification at several time points, particularly in the lymph nodes. However, although there were no significant changes in the number or diversity (measured as Shannon Entropy; Sh) of barcodes before and after each galunisertib cycle (Supplementary Fig. S16A and Fig. 8A), we found significant changes in barcode frequency distribution in most tissues and cycles after galunisertib treatment (Fig. 8B and

Supplementary Fig. S16B). Specifically, the relative proportion of each barcode changed in all monkeys in all cycles in the rectal biopsies (probably a consequence of different sampling area), but also for all LN analyzed (except for 08M134 in cycle 1). Of note, the same LN were sampled at the beginning and after cycle 3 (Supplementary Table S3). Hence, sampling location does not explain the changes barcode distribution. Changes in the proportion of barcodes were also detected in at least half of the PBMCs after cycle 1 and 2 (cycle 3 not analyzed).

Finally, barcode diversity decreased in plasma post-ATI compared to the time point right before ART (Fig. 8C) and barcode distribution significantly changed post-ATI compared to pre-ART in 2 of the 7 macaques that rebounded (Supplementary Fig. S16C).

## TCF1 downregulation in CD4+ T cells correlates with virological and immunological endpoints

In order to explore a possible association between the various virological and immunological parameters and their changes, we built a correlation matrix with a curated set of variables of interest or their fold changes. This analysis revealed an association between the decrease in TCF1 expression in the CD4+ T cells and several virological and immunological variables (Fig. 8E). Specifically, both the levels of CA-vDNA in the colorectal tissue and the fold increase in gut-SUV strongly inversely correlated with fold changes in TCF1. Since TCF1 decreased, a larger decrease in TCF1 was directly proportional to residual vDNA in the gut at the end of cycle 4 and to the increase in PET signal in the gut (Fig. 8D, E). A weaker, but still significant association was also present with CA-vDNA in PBMC at the end of cycle 3 and with the levels of vRNA in the lymph nodes at the end of the 4th cycle (Fig. 8D and Supplementary Fig. S17A). Finally, the decrease in TCF1 correlated with the increase in IFN-γ and poly-functional IFN-γ/TNF-α releasing cells (cumulative increase of SIV-specific responses to gag, pol and env; Fig. 8D, E). Of note, the increase in PET signal in the gut also correlated with the levels of vDNA in PBMC at the end of cycle 3, vDNA in gut biopsies and residual CA-vRNA in LN at the end of cycle 4 (Supplementary Fig. S17B). Interestingly, the levels of residual CA-vRNA in LN also correlated with the increase in IFN-γ and poly-functional IFN-γ/TNF-α releasing cells (Pearson r = 0.85 and 0.76, respectively). Finally, the residual intact pro-virus (IPDA) directly correlated with the levels of IFN-γ and IFN-γ/TNF-α produced in response to SIV peptides at the end of cycle 3 (Fig. 8D; r = 0.76 and 0.88, respectively). In contrast, the change in intact provirus trented to correlate inversely with the levels of IFN-γ, so that a larger decrease directly correlated with more IFN−γ responses. However, this did not reach significance (p = 0.117, Supplementary Fig. S17C).

## Discussion

HIV-1 latency in T cells is maintained through diverse mechanisms that include blocks in transcriptional elongation, completion, and splicing[13]. A common characteristic shared by HIV-1 latently infected cells of both T and myeloid cell lineages is their "resting" phenotype[14,54-56] In these cells, an inability to transcribe proviral DNA is linked to a generalized decrease in transcriptional activity which, in turn, is linked to their metabolic status[23]. Cellular metabolism is in turn influenced by tissue location and environmental cues[57].

TGF-β is released at high level in PLWH and its levels remain high during ART[58-60]. The immunosuppressive activity of TGF-β is well-known. However, the effect of TGF-β signaling in immune cells is highly context dependent[61]. Hence, TGF-β plays different roles according to a cell differentiation and activation status[61]. In CD8+ T cells and NK cells, TGF-β was shown to decrease mTOR activity and preserve cellular metabolism (high mitochondrial activity and spare respiratory capacity, but reduced mTOR activity) preventing metabolic exhaustion[25,26,62,63]. This effect was linked to survival of antigen-specific CD8+ T cells, preservation of their stemness and it was linked to higher expression of the TCF1 factor[25].

In contrast, in CD4+ T cells TGF-β is known to decrease TCR activation[29,64], restrict proliferation and inhibit cytotoxicity (including granzyme and perforin release) at different stage of infection in vivo[30,65]. However, the role of TGF-β in the formation and preservation of CD4+ T cell memory is still unclear[61]. Moreover, the link between TGF-β signaling and TCF1 expression in CD4+ T cells is unexplored.

Here, we used a clinical stage small drug, galunisertib, developed by Eli Lilly and used in several phase 1 and 1/2 clinical studies against solid cancer[39,66,67] to investigate the impact of TGF-β blockade on SIV latency, SIV reservoir and immune responses. Of note, Eli Lilly did not terminate galunisertib development program because of toxicity[68,69]. Indeed, in our studies in macaques, we observed no adverse events nor changes in chemical or hematological variables. Moreover, there were no detectable changes in the levels of the 24 inflammatory factors that we probed in plasma during the treatment. This suggests that this therapeutic approach may be safe in people living with HIV (PLWH).

The first important finding of our study was the confirmation of our previous report of the latency reversal properties of TGF-β blockade in vivo[35]. Indeed, we found increase in pVL in 7 out of the 8 macaques upon initiation of galunisertib therapy. Although not all macaques were fully suppressed before treatment, substantial increases in pVL (>10^2 copies/mL) were noted also in fully suppressed macaques (08M171, A8R095 and A8L057). Moreover, viral reactivation was documented in tissues by immunoPET/CT. Importantly, the SUV increase detected post-galunisertib in gut and LN correlated with CA-vRNA as in our previous studies[35] and, interestingly, it was associated with a decrease in TCF1. However, despite this evidence, the absence of imaging studies carried out in uninfected, galunisertib-treated macaques require that we interpret this data with care. This is due mostly to unexpected and, yet unexplained, galunisertib-driven changes in BPA. Without a better understanding of these changes, it is difficult to determine the BPA contribution to the PET signal increase in tissues. Yet, an increase in gut and LN SUV is present even after BPA normalization (although in cycle 1 instead than cycle 3). Since our probe is a F(ab')₂ and not a whole antibody, we did not expect the probe to be still present in significant amounts in circulation in a scan performed 24hrs post-probe injection. Nonetheless, galunisertib may have impacted probe and probe-antigen complex pharmacokinetics or the probe interaction with increased viral antigen. TGF-β is required for vascular barrier function[70]. Hence, galunisertib may have increased vascular permeability. However, this would have driven a major decrease in BPA instead than the detected increase. Moreover, significant changes in VEGF-A, a factor critical to and tracking with vascular permeability[71], were not noted. This, in conjunction with our previous studies which validated the specificity of the PET signal for areas of enhanced SIV replication in gut and lymph nodes[35,43,72], demonstrates that the galunisertib-driven increases in SUV were likely specific, and identified areas of SIV latency reversal at least in gut and lymphoid tissues. The extent to which the increased signal in the spine and bones recapitulates an increase in SIV replication at these sites remains to be determined.

Importantly, we observed a decrease in CA-vDNA in all the tissues that cannot be attributed to ART alone. Indeed, although we did not have a concurrent control group, this decrease was not present in similar studies conducted in SIVmac239M2 infected macaques on the same ART regimen, but not treated with galunisertib. Considering studies by other groups with different models (SIVmac251), they report 2nd phase decay of SIV intact provirus (weeks -32 to -100pi, Fig. 2B in ref. 73) with a t₁/₂ of >8 months[73]. In contrast, in our study, the intact provirus decreased by 3 fold (median) in a little over 3 months (from week 35pi, BC1 to week 49pi, AC4). Interestingly, this decrease in intact pro-virus trended toward a direct correlation with IFN-γ responses. However, IFN−γ levels also inversely correlated with the absolute value of residual intact provirus, suggesting that IFN-γ responses were driven by residual viral reservoir while, at the same time, were involved in clearing intact virus. Indeed, the increase in IFN-γ and TNF-α also correlated with residual CA-vRNA in the LN at the end of the treatment. The latter, in turn, was directly proportional to the increase in gut SUV. This is in line with increased latency reversal explaining residual viral RNA in lymphoid tissues.

Finally, one of our most intriguing results was the profound downregulation of TCF1 in CD4+ T cells at both the transcriptional and protein levels. Although TCF1 is conventionally viewed as an effector of

the canonical Wnt pathway[74] and recently reached notoriety for its role in maintaining stemness of antigen-specific memory CD8+ T cells[26,75], TCF1 has a plethora of functions in T cell development and differentiation largely independent of Wnt signaling[22]. In CD4+ T cells, TCF1 has been implicated in orchestrating all the major Th subsets, including Th1, Th2, Th17 and Tfh[76]. TCF1 is known to control the bifurcation between Th1 and Tfh in favor of Tfh cells[76], while it negatively regulates Treg development[22]. This is in line with our findings of increased Treg and decreased Tfh, in the midst of a profound downregulation of TCF1. Importantly, TCF1 is downregulated with increased cellular differentiation and progression toward effector functions in T cells. T cell activation leads to reduced levels of TCF1 and higher levels of TCF1 are present in T cells with higher stemness and low anabolic metabolism[74]. These findings suggest that TCF1 has a critical role in maintaining quiescence in immune cells likely in concert with TGF-β[18,65]. Our data reveal that this link may be even more prominent and important in CD4+ T cells than in CD8+ T cells. Importantly, the decrease in TCF1 was accompanied with enhancement in other effector markers such as CD95, CD16 and GRZB (at the transcriptional level) and an increase in the transcription of AP1 complex. However, there was no clear upregulation of other classical markers of immune activation such as CD69 and no increase in T cell proliferation (Ki67 expression). Hence, galunisertib treatment does not appear to lead to classical T cell activation nor to an increase in a specific terminally differentiated effector subset. Instead, in vivo TGF-β blockade seems to primarily change the metabolic state of T cells (and likely other immune cells) increasing OXPHOS and mitochondrial function. Of note, although glycolysis is essential during cell activation, mitochondrial pathways are engaged and remodeled early after activation and OXPHOS upregulation has a pivotal role in the earliest stages of cell activation[18].

Hence, we propose a model in which TGF-β inhibition forces cells (particularly CD4+ T cells) out of quiescence to a transitional state where they reinitiate their transcriptional program and are metabolically ready to be activated. Because of the highly context-dependent effect of TGF-β, the final impact of galunisertib is likely heterogenous and dependent on other intrinsic and extrinsic cellular stimuli. In absence of direct TCR engagement or other activation stimuli, the majority of the T cells do not undergo full/classical activation and proliferation following galunisertib treatment. Instead, the cells are pushed toward a more effector-like phenotype. This explains our observation of an enrichment of transient effector or "transitional effector" T cells that, in turn, can reinitiate viral transcription and more promptly respond to antigenic stimulation. Indeed, functionally, we demonstrated that the PBMC after galunisertib treatment secrete higher levels of IFN-γ and TNF-α. Interestingly, there was no increase in IL-2 secretion. The link between TGF-β and IL-2[77] and the critical role of IL-2 in T cell proliferation again suggest that galunisertib enhances an effector phenotype uncoupled from cellular proliferation. scRNAseq analysis demonstrated a trend toward an effector phenotype also in other immune cells, such as B cells, NKs and macrophages. Future studies will need to uncover in depth the effect of TGF-β blockade on these other immune subsets.

This study has several limitations. The most important limitations are the relatively small number of macaques and the lack of a concurrent untreated control group. We also could not investigate in depth the viral kinetics after ART interruption because of the short follow up after ATI. An additional important limitation is the lack of immunoPET/CT images from an uninfected control group of macaques treated with galunisertib. This control group may have given us insight on the impact of galunisertib on the pharmacokinetics of the immunoPET/CT probe in absence of antigen. Moreover, we did not explore changes in the phenotype or turnover of cells isolated from gut and lymph nodes and relied solely on transcriptional data for these tissues. Although we found an association between TCF-1 downregulation, enhanced effector function (IFN-γ release) and measures of latency reversal, a causal link

between increased effector phenotype and latency reversal was not definitively established. Finally, because of sample availability, we could not dissect the cellular origin of increased IFN-γ and TNF-α.

In conclusion, we report that in vivo treatment with a clinical stage small molecule TGF-β inhibitor drives a transitional effector phenotype in T cells that is likely responsible for increasing the frequency of spontaneous latency reversal events, stimulating SIV-specific immune responses, and decreasing the viral reservoir. Future work will determine whether the galunisertib-driven enhanced antiviral responses and decreased viral reservoirs can significantly contribute to post-ART virological control.

## Methods

### Study design and Ethics Statement

A total of 8 adult female Indian-origin *Rhesus* macaques (*Macaca mulatta*; Mamu A*01, B*08 and B*17 negative) were used for the study described in the manuscript (Supplementary Table S1). All the macaques were selected from the colonies bred and raised at the New Iberia Research Center (NIRC), University of Louisiana at Lafayette. All animal experiments were conducted following guidelines established by the Animal Welfare Act and the NIH for housing and care of laboratory animals and performed in accordance with institutional regulations after review and approval by the Institutional Animal Care and Usage Committees (IACUC) of the University of Louisiana at Lafayette (2021-8821-002; protocol 8821-01).

Rhesus macaques ($n = 8$ main study +4 separate non-concomitant study) were infected with 300 TCID$_{50}$ of the barcoded SIVmac239M2 stock intravenously and ART (Tenofovir [PMPA] at 20 mg/ml, Emtricitabine [FTC] at 40 mg/ml and Dolutegravir [DTG] at 2.5 mg/ml) was initiated on week 6 pi. Galunisertib treatment was initiated on week 35 p.i. Powder (MedChemExpress – MCE, NJ, USA) was dissolved in water and given orally in a treat twice daily at 20 mg/kg. 4 cycles of 2 weeks daily treatment with 2 weeks wash out period were performed. Macaques 08M156 and A6X003 were given the rhesus recombinant antibody (rhesus/human chimeric) anti-PD1 antibody [NIVOR4LALA; comprising silenced rhesus IgG4k constant regions and variable regions from anti-human PD-1, nivolumab; non-human primates reagents resource, NHPRR; 5 mg/kg] at the beginning of the 3rd and 4th galunisertib cycles.

Blood viral load was monitored biweekly before and during ART and every 3-4 days during Galunisertib treatment. Colorectal biopsies and LN FNA were collected before and after galunisertib treatment. ART was terminated 3 weeks after the last galunisertib dose, and euthanasia and necropsy to harvest tissues were performed at week 58 post infection. Tissue samples were flash frozen, fixed in OCT or Z-fix.

### Plasma and Tissue SIV Viral loads (VL)

Blood was collected in EDTA tubes, and plasma separated by density gradient centrifugation was used for the determination of plasma VL by SIVgag qRT-PCR at NIRC or at Leidos (Quantitative Molecular Diagnostics Core, AIDS and Cancer Virus Program Frederick National Laboratory). Tissue VL from snap-frozen PBMC pellets, colorectal biopsies and LN FNA were quantified as described in ref. 78. Briefly, tissue viral DNA and RNA loads were measured, respectively, by qPCR and qRT-PCR with standard curve method and normalized to Albumin copy number (for cell-associated viral DNA) and total RNA quantity. DNA and RNA were extracted from snap-frozen tissues using DNeasy/RNeasy blood and tissue kits (Qiagen) following the manufacturer's instructions. Primers: SIVgag FW (5′-GGTTGCACCCCCTATGACAT-3′), SIVgag RV (5′-TGCATAGCCGCTTGATGGT-3′), SIVProbe (5′-6-FAM-AAT CAGATGTTAAATTGTGTGGGA-3′); macaque Albumin FW (5′-ATTTT CAGCTTCGCGTCTTTTG-3′), RV (5′-TTCTCGCTTACTGGCGTTTTCT-3′), Probe: (5′-6-FAM-CCTGTTCTTTAGCTGTCCGTG-3′). SIV-IPDA was performed on freshly stored PBMC before cycle 1 and at the end of cycle 4 of galunisertib by Accelevir, Baltimore, MD.

## ImmunoPET/CT

ImmunoPET/CT for mapping SIV signals in total body scans were conducted in part as reported[44]. The probe consisted of primatized p7D3 anti-env F(ab)'2 coupled with the chelator DOTA and labeled with $Cu^{64}$ just prior to administration to the animals. For probe administration, the animals were sedated, and a venous catheter was placed into an arm of leg vein to minimize bleeding of the probe into the tissue surrounding the site of injection. The probe for each animal consisted of ~1 mg of the p7D3 F(ab)'2 labeled with 2-3 mCi of $^{64}Cu$. After injection, the animals were allowed to recuperate in their cage until the next day. At 24 h, the animals were re-anesthetized with Telazol and immobilized in dorsal recumbency on the scanner table. Scans were conducted in a Phillips Gemini TF64 scanner. The final CT image was compiled from 200 to 300 slices, depending on macaque's size.

PET Image analysis was performed using the MIM software. PET/CT fusions were generated and scaled according to calculated Standardized Uptake Values (SUV). The SUV scale for the PET scans was selected based on the overall signal intensity of the PET scans (whole body), and the CT scale was selected for optimal visibility of the tissues. All images and maximum image projections (MIP) were set to the same 0-1.5 scale for visual comparisons. Additional details on MIM analysis are described in Supplemental Methods.

## Cell isolation, flow cytometry staining, classical and high-dimensional analysis

Colorectal biopsy tissues were isolated by enzymatic digestion, while LN biopsies were passed through a 70 µm cell strainer as described in[44]. Isolated cells were phenotyped with panels listed in Supplementary Table S2. FlowJo V10.8 was used for both classical and high-dimesional analysis. PeacoQC, FlowtSNE, and FlowSOM plug-ins were used with default settings. Normalization was performed using SwiftReg[47]. More details on the staining procedures and analysis pipeline can be found in Supplemental Methods.

## Bulk and scRNAseq analysis

For bulk RNAseq, snap-frozen PBMC pellets from BC1, 1 h after the first Gal dose, and AC1 were used for RNA extraction with the RNeasy kit with on-column DNA digestion (Qiagen). Library preparation was performed using TruSeq Stranded Total RNA with Ribo-Zero Globin, and sequencing was done with an Illumina HiSeq4000 with >20 M reads/sample. Sequencing data were demultiplexed and trimmed using Trimmomatic v0.36 to remove adapters and low-quality reads. Trimmed reads were aligned to the Mmul10 reference genome, and transcripts were quantified using the Hisat2-StringTie pipeline[79]. Differential gene expression analysis using the quantified gene transcripts was performed with DESeq2 R package[80], comparing the samples attained before and after galunisertib treatment and controlling for intra-animal autocorrelation. Differentially expressed genes (DEGs) were analyzed by functional enrichment analysis and gene set enrichment analysis (GSEA) to identify specific pathways and molecular processes altered by galunisertib.

The Parse pipeline and Partek software were used for scRNAseq analysis. For bulk RNAseq, features with <100 counts were removed, data were normalized, and DESeq2 was used to obtain a DEG list. Genes with a false discovery rate (FDR)-adjusted $p$-value ≤ 0.05 and absolute $\log_2$ fold-change (FC) (compared to BC1) above 2 were defined as significantly differentially expressed (DEG). For scRNAseq analysis, cells isolated from lymph nodes before (BC1) and after (AC1) galunisertib were fixed with the Parse fixation kit, barcoded, and sequenced at the NUseq Core. The Partek software was used for scRNAseq analysis. Cells with 400-8000 features, excluding features with 0 reads in >99.99% cells, were included. Scran deconvolution was used for normalization and cell classification was based on gene expression. Hurdle models were used to compare DEGs in each cell subset before and after galunisertib. See supplemental methods for a detailed description of

scRNAseq analysis and additional control analysis using Seurat R package with more stringent QC cut-offs and SCTransform normalization.

## Plasma cytokines and T cell responses

Cytokines in plasma (at 1:2 dilution) were measured using the NHP Cytokine 24-Plex kit by Meso Scale Diagnostics (MSD) according to manufacturer instructions. Frozen PBMC collected at week 6 post-infection (pre-ART), right before the first galunisertib administration (BC1), at the end of cycle 1 (AC1), and at the end of cycle 3 (AC3) were thawed in AIM V medium (Thermo Fisher) with benzonase (Sigma) and plated on a FluoroSpot (CTL) plate pre-activated with 70% Ethanol and IFN-γ, TNF-α and IL-2 capture solution. Gag, pol, and env 15-mer peptides (NIH AIDS Reagents program) were prepared at two times the final concentration of 2.5 µg/mL with co-stimulatory reagents anti-CD28 10 µg/mL and anti-CD49d 10 µg/mL and added to the cells in CTL-Test™ Medium. Parallel positive control of PMA (20 ng/mL) and ionomycin (200 ng/mL), or mock DMSO solution, was also plated with the stimulatory reagents. PBMCs were added at 300,000 cells per well. After 24hrs, the plate was washed, incubated with detection and tertiary solutions, and shipped to CTL for scanning and QC.

## Statistics

GraphPad Prism v10, R and Python were used for statistical analysis and data visualization. Wilcoxon matched-pairs test and repeated measures ANOVA or mixed-effect analysis (when the dataset had missing data) were used to compare the different virological and immunological variables between baseline (BC1) and a single or multiple post-galunisertib time points. In Prism, the mixed model uses a compound symmetry covariance matrix and is fit using Restricted Maximum Likelihood (REML). Cytokine data from MSD assay were first Log-transformed, then normalized by subcolumn/factor in percentage with 0% as the smallest value and 100% as the larger value in each dataset. Each factor was analyzed separately with ANOVA for repeated measures and Dunn's multiple comparison's post-hoc test and together by principal component analysis (PCA). The Holm-Sidak test was used for multiple comparison correction in all cases, but FlowSOM cluster comparison where the FDR method was used. Pearson r coefficient was calculated for pairwise tests of association in a correlation matrix with selected variables. For RNAseq analysis see above and supplemental methods. For viral population analysis, Shannon Entropy of the viral barcodes present in each sample was used to measure the diversity of the viral populations (see supplemental Methods). Chi-squared tests were used in R to compare barcode composition between before and after cycle time points within a tissue/macaque using paired barcode relative frequencies. Unless otherwise specified, $p$-value < 0.05 was considered statistically significant.

## Reporting summary

Further information on research design is available in the Nature Portfolio Reporting Summary linked to this article.

# Data availability

All relevant data are included in the manuscript or supplemental material. Source data are provided with this paper in the Source Data file. Raw data files including DICOM image files are available to be shared upon request to the corresponding author elena.martinelli@northwestern.edu. All RNA sequencing data originating from this study have been deposited in NCBI GEO under the accession code: GSE244871. Source data are provided with this paper.

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

## Acknowledgements

General: We acknowledge the helpful staff at the New Iberia Research Center as well as the help of Dr. Suchitra Swaminathan and the rest of the staff of the Flow Cytometry Core Facility at the Robert H. Lurie Comprehensive Cancer Center of Northwestern University in Chicago. Moreover, we thank the staff of the NUseq Core at Northwestern, and in particular Dr. Ching Man Wai and Matthew Schipma for the help with RNAseq data. Finally, we thank Dr. Lifson and the Leidos staff for viral load measurements as well as Dr. Laird at Accelevir for providing SIV-TILDA results in a timely manner. The content of this publication does not necessarily reflect the views or policies of the Department of Health and Human Services, nor does mention of trade names, commercial products, or organizations imply endorsement by the U.S. Government. Funding: This project has been funded by National Institutes of Health grant R56AI157822 and R01AI-176599 to Dr. Martinelli, the resource for NHP immune reagents to Dr. Villinger (R24 OD010947) for probe productions and in part with federal funds from the National Cancer Institute, National Institutes of Health, under Contract No. 75N91019D00024/HHSN261201500003I. The Lurie

Cancer Center is supported in part by an NCI Cancer Center Support Grant #P30 CA060553.

## Author contributions

EM conceptualized the studies; JK performed and analyzed flow cytometry and immunePET/CT data; DB, MA, DF and SA collected and processed macaque samples; MRH generated vRNA/DNA and Elispot data; RC and MV analyzed Treg/Tfh; YT and TJH contributed to immunePET/CT analysis; EG, YG and EM analyzed RNAseq data; CMF, BK and RLR analyzed barcode data; JA, CC, FV and EM interpreted the data and wrote the manuscript.

## Competing interests

The corresponding author's institution, Northwestern University filed a patent application including all the data from the present manuscript. Application number: 18/515,196, Filing date: November 20, 2023. Inventor, Elena Martinelli. All other authors declare no competing interests.
