## [Peer Review File · Nature Communications]

TGF- β blockade drives a transitional effector phenotype in T cells reversing SIV latency and decreasing SIV reservoirs in vivoEditorial Note: The Peer Review File has been amended from the original version to redact the name of a researcher.

Reviewers' Comments:

Reviewer #1:

Remarks to the Author:

Kim et al. perform further analyses on 8 indian rhesus macaques and 4 historical 'control' on the effects of the TGFbeta inhibitor, galunisertib as a latency reversal agent during early SIV infection. They demonstrate the potential for galunisertib as a therapeutic to reactivate latency resulting in the potential to reduce SIV reservoirs associated with enhanced effector function in NK and T cells and reduced Tregs and Tfh. In addition, there is enhancement of OXPHOS transcriptional pathways suggesting improved memory function. This also translated to enhanced SIV specific T cell immunity after multiple doses of TGF beta inhibition. These findings suggest novel therapeutics in cure treatment regimens that should be tested in humans.

One concern that is mentioned in the manuscript is the lack of proper control group for these analyses.

Fig S4 shows data from 4 historical control macaques showing ca-vDNA at weeks 24 and 52.

In 25% of these subject, ie 1 / 4, the DNA goes down. One should show a similar plot of the 8 tested subjects as comparisons in the graph and determine if there is any statistical significance, although it might be difficult with only four controls. Was intact provirus determined in the four control subjects. In addition, regarding intact provirus in the treated subject, 2/8 showed no change. so 25% of treated subject showed no change, but 75% of controls showed no change- can a fisher's exact test be done?

In the Discussion they suggest that elevated TGFbeta is common in ART treated individual, however, I do not see a specific study or reference finding this when compared to HIV negative individuals. Please include this reference.

Reviewer #2:

Remarks to the Author:

This is a well-presented manuscript that provides significant data on the SIV latency reversal properties of a TGF-beta inhibitor, galunisertib, and its downstream effect on viral reservoirs and immune response. The authors collected substantial data from 8 SIV infected rhesus macaques that received 4 cycles of galunisertib treatment. Samples were collected before and after each cycle to show the progressive changes as a result of TGF-beta blockade. The tissue and blood data strongly support the conclusions presented in the manuscript. This is original with highly significant findings.

Comments:

- 1) The rationale for administering anti-PD1 in two of the primates was not well articulated.
- 2) There are several issues with the imaging data that must be addressed before further consideration of this manuscript.
 - i. An additional limitation of this study was the absence of a control group for the PET imaging component. Uninfected rhesus given the four dose TGFb regimen would have been a suitable control to assess any pharmacokinetic changes in ^{64}Cu -p7D3 fragment distribution as a function of the treatment. While, understandably, the antigen may not be present in the uninfected primates, there may be TGFb treatment specific pharmacokinetic changes. Given that the viral load in the blood remains low over the treatment course (Figure 1B), the increase in SUV noted in cycle 2 and 3 in some of the primates is confounding.
 - ii. The color scale bar in Figure 2A does not have any units associated with it. Is this SUV? The scale is very compressed such that small changes will be manifested in intense color changes on the images. Are the images in 2A and S1 maximum intensity projections (MIPs) or coronal slices?

- iii. The Y-axis units in Figure 2B are unfamiliar – Total SUV. Typically, image analysis is presented as SUVmax or SUVmean. Total SUV is not a measure that is used in PET imaging analysis. Were the SUVs corrected for the blood pool activity?
- iv. It is unclear how the regions of interest were defined especially for the gut analysis. One has to be careful to avoid spillover from other organs and radioactive stool will muddle the analysis. Axial images especially of the lymph node regions showing the ROIs would be helpful for review and add to the data pool.
- v. Where does the whole-body increase (figure S2) in signal come in AC2, BC3 and AC3 if all of the primates are injected with nearly the same amount of activity for each scan? Again SUV total is not a typical measure used for image analysis.
- vi. Were other correlative studies performed on tissue viral load (e.g. lymph nodes, marrow), by biopsy, and uptake of the ⁶⁴Cu agent as a function of TGFβ inhibition over the 4 cycles?
- vii. The correlation of the viral load in colorectal biopsies and the SUV (Figure S3) is not very strong. This does not seem to match the image data. Also the description of the gut analysis: Gut abdomen ROI – organs is unorthodox and may not account for stool in the intestines. Also, the vRNA concentration is typically reported in the HIV literature on the log scale. All the data in S3 is linear and not large changes in concentration over the course of treatment.
- viii. There is a disconnect between the later images for some of the primates showing increased activity throughout the body and the viral load in blood and tissues.
- ix. The authors are strongly encouraged to have the PET imaging data thoroughly re-analyzed by a medical physicist or scientist with experience in PET image analysis and modeling.

In general the authors need to decide between keeping the imaging data in this manuscript, after considerable reanalysis, or remove it and publish with the remaining overall strong data. In the current state the imaging data does not strengthen the manuscript and may even detract from the strength of the biological data

Edits:

Page 4 line 103. Remove "s" from "T cells"

Throughout manuscript and figures. ml should be mL

Page 17 line 425 Remove "s" from "3 folds"

Reviewer #3:

Remarks to the Author:

This manuscript by Kim et al represents a clear contribution to our knowledge of cytokines that influence the ability of HIV to remain latent in non-replicating T cell subsets. These studies in the SIV-macaque model represent an important step toward our ability to test TGF-β blockade as a latency reversal agent in humans.

One area of concern is with regard to the ability of the authors to determine that it was an increase in effector or transitional effector T cells that was the major contributor in reversing the latency phenotype. Authors need to more clearly discuss the evidence for this in the results/discussion to make a compelling case that this is the key mechanism for the observed alterations in viral levels. Also, authors need to more clearly discuss the use of the word 'transitional' in the title. The only place where they use this word in the results/discussion is on line 458, where they hypothesize that the cells are in the beginning of the effector stage. Outlining the evidence and a better definition for what they mean by transitional would be useful in understanding the relationship between the increased viral expression with this altered T cell phenotype (which appears to be primarily described as TCF1 low in the manuscript).

Minor issues:

Methods: concentrations of ART drugs needs to be added. Anti-PD1 antibody concentration utilized

needs to be added.

Abbreviations: IPDA abbreviation not explained during first usage

Figure 4G. top label is missing from some of the plots.

4H and 4I: need to better explain how the populations were derived (supplemental figure) and to identify important populations in the figure legend.

Line 311: there is no figure 6J

Figure 7. TNF alpha appears to be increased similarly irrespective of peptide, and in the absence of peptide, to a similar extent. Text should reflect that the T cells are more TNF-alpha producing in general. Unless there is a statistical difference between gag peptides and DMSO that is not stated or obvious. Current sentence is unclear in this regard (Lines 332-333).

Reviewer #1 (Remarks to the Author):

Kim et al. perform further analyses on 8 indian rhesus macaques and 4 historical ‘control’ on the effects of the TGFbeta inhibitor, galunisertib as a latency reversal agent during early SIV infection. They demonstrate the potential for galunisertib as a therapeutic to reactivate latency resulting in the potential to reduce SIV reservoirs associated with enhanced effector function in NK and T cells and reduced Tregs and Tfh. In addition, there is enhancement of OXPBOS transcriptional pathways suggesting improved memory function. This also translated to enhanced SIV specific T cell immunity after multiple doses of TGF beta inhibition. These findings suggest novel therapeutics in cure treatment regimens that should be tested in humans.

One concern that is mentioned in the manuscript is the lack of proper control group for these analyses. Fig S4 shows data from 4 historical control macaques showing ca-vDNA at weeks 24 (28) and 52. In 25% of these subject, ie 1 / 4, the DNA goes down. One should show a similar plot of the 8 tested subjects as comparisons in the graph and determine if there is any statistical significance, although it might be difficult with only four controls. Was intact provirus determined in the four control subjects. In addition, regarding intact provirus in the treated subject, 2/8 showed no change.so 25% of treated subject showed no change, but 75% of controls showed no change- can a fisher’s exact test be done?

We appreciate the important point raised by the reviewer. We did not run the intact provirus IPDA assay on the control samples, because we did not have the frozen cells necessary for this assay at the week 52 time point. However, to address this reviewer’s concern, we ran the IPDA on the time points we had left: weeks 28/32 vs week 38. As shown in the graphs below at this stage of the infection intact provirus decreased in 2 animals and increased in the other 2. Directly comparing these data with the data shown in our study would be incorrect considering the relatively large difference in the time period (weeks 28/32 vs 38 in controls and weeks 35 vs 49 in the study – see below). The control group also had lower intact provirus than our study group at baseline. This said, we could compare the CA-vDNA data in the manuscript between study group (Fig 3A) and controls (Fig S4B) with the Fisher’s exact test (8/8 decrease in study group vs 1 / 3 decrease in the controls) and it would be significant. However, the new IPDA data rather suggests that we should use more caution in comparing these different non concomitant studies. Hence, we decided to further highlight the limitation of a lack of proper control group and soften our conclusion regarding the relative effect of galunisertib and ART on the viral reservoir (lines 224- 229).

In the Discussion they suggest that elevated TGFbeta is common in ART treated individual, however, I do not see a specific study or reference finding this when compared to HIV negative individuals. Please include this reference.

We had 1 reference which was a review. We took out that reference and added 3 non-review references that include comparisons with uninfected individuals (Line 480)

Reviewer #2 (Remarks to the Author):

This is a well-presented manuscript that provides significant data on the SIV latency reversal properties of a TGF-beta inhibitor, galunisertib, and its downstream effect on viral reservoirs and immune response. The authors collected substantial data from 8 SIV infected rhesus macaques that received 4 cycles of galunisertib treatment. Samples were collected before and after each cycle to show the progressive changes as a result of TGF-beta blockade. The tissue and blood data strongly support the conclusions presented in the manuscript. This is original with highly significant findings.

Comments:

1) The rationale for administering anti-PD1 in two of the primates was not well articulated.

The rationale for the combination originated from the demonstrated synergistic activity of anti-PD1 and TGFb-blockade in cancer models and clinical trials including an increased stimulation of anti-cancer immunity over ICI therapy alone. A sentence in this regard was added in the text (lines 139-140).

2) There are several issues with the imaging data that must be addressed before further consideration of this manuscript.

i. An additional limitation of this study was the absence of a control group for the PET imaging component. Uninfected rhesus given the four dose TGFb regimen would have been a suitable control to assess any pharmacokinetic changes in ^{64}Cu -p7D3 fragment distribution as a function of the treatment. While, understandably, the antigen may not be present in the uninfected primates, there may be TGFb treatment specific pharmacokinetic changes. Given that the viral load in the blood remains low over the treatment course (Figure 1B), the increase in SUV noted in cycle 2 and 3 in some of the primates is confounding.

We appreciate the reviewers' concerns. Data analysis and interpretation of these images is complicated by the relatively novel use of immunoPET/CT to measure viral burden in tissues. In this regard, a first important point to consider is that viral burden in blood (aka plasma viral load) is not expected to reflect viral burden in tissues. The power of using immunoPET/CT lies in revealing host-viral dynamics in tissues (where viral replication is often focal in nature) that remain hidden when only studying the blood. Indeed, several groups have now reported increases in PET signal upon early infection/ART interruption in absence of or preceding plasma viral load detection (Obregno-Perko JCI Insight 2021; Hope TJ HIV Persistence Workshop 2019 & 2022). Hence, it is somewhat expected that the images in the current manuscript do not exactly recapitulate the plasma viral load dynamics.

This said, we agree with the reviewer that having control images from uninfected macaques treated with the same therapeutic regiment as in the infected macaques in our study would provide additional insights into galunisertib-driven changes in the pharmacokinetics of our probe. We have discussed this limitation in the text (lines 509-510 and 600-604). Indeed, we are looking into ways to gather the resources to perform such lengthy and expensive control studies. However, as implied by the reviewer, even these controls may not fully capture changes that may be due to the drug impact on probe-antigen dynamics. Hence, we followed the reviewer's suggestion to consult with a radiologist expert [redacted] and re-analyzed the data following the expert's suggestions. Ultimately, even after checking our contouring strategies and providing additional measurement and normalizations, and after addressing all the concerns raised by the reviewer below, we believe that the imaging data contributes to the overall study, constitutes an important and novel way to gather insights on the effect of the drug, and should remain in the manuscript.

ii. The color scale bar in Figure 2A does not have any units associated with it. Is this SUV? The scale is very compressed such that small changes will be manifested in intense color changes on the images. Are the images in 2A and S1 maximum intensity projections (MIPs) or coronal slices?

We thank the reviewer for highlighting the missing information (SUV and MIP). They have now been added to the manuscript where needed.

iii. The Y-axis units in Figure 2B are unfamiliar – Total SUV. Typically, image analysis is presented as SUV_{max} or SUV_{mean}. Total SUV is not a measure that is used in PET imaging analysis. Were the SUVs corrected for the blood pool activity?

SUV_{max}, SUV_{mean} and SUV Total were all evaluated and compared in our early studies (Santangelo et al. Nat Methods 2015) to determine the variable that would best describe the differences between infected, uninfected and controller animals. While SUV_{max} is a critical value in oncology studies, our initial studies demonstrated that this measure was less useful when attempting to quantitate an infection in tissues. Most notably, SUV_{max} was subject to variations of signal across and within specific organs and, ultimately, proved to be a poor way to compare signals detected in SIV controller vs non-controllers. Instead, we concluded that SUV_{mean} and SUV Total provided better measurements for our experimental system, representing, respectively, the mean and total signal across a tissue ROI or organ. We decided to use only the SUV Total in the first version of our manuscript because it had also been used by other groups in similar studies (Obregno-Perko JCI Insight 2021). After consulting with our nuclear medicine expert, we decided instead to include the SUV mean in the main figure (see revised Fig 2). We also considered using the Total Lesion Glycolysis (TLG) normalized by body weight in place of SUV total to represent the total amount of radiotracer in the ROI or, alternatively, to use the %ID/organ (percentage of injected dose). However, TLG and %ID/organ are very similar to each other for full organs and they both are highly dependent on the ROI volume. Hence a minimal change in ROI may result in large and potentially misleading differences in TLG or %ID. Because of the way we draw the ROI (maintaining as much as possible similar ROIs across time points for each macaque), the SUV Total is a better parameter than TLG to compare total activity in a specific ROI across time points. We added this measure in the supplementary material (Fig S3A).

Regarding the blood pool activity (BPA), we initially did not include it in our analysis, because in our initial studies (Santangelo et al. Nat Meth 2016) with a whole 7D3 antibody as probe, we observed minimal blood pool activity when imaging 24hrs after probe injection. Since our current probe includes only the F(ab')₂ portion, which has faster clearance from the bloodstream, we thought the blood pool activity would be negligible. Indeed, it was negligible in our previous published (Santangelo 2016, Samer 2021) and unpublished studies. However, this was an oversight in the case of the present study. Upon reanalysis of the data, we measured the blood pool activity at the level of the left ventricle cavity, and we found some significant changes that may be due to the drug impact on the probe or on the probe/antigen interaction (Fig S3B). These changes are difficult to interpret. We hope to share the raw images with the scientific community in an appropriate repository or upon demand for reanalysis to foster the debate. In light of the known critical role of TGF- β in the establishment and maintenance of endothelial barrier function (Walshe et al 2009 among others), TGF- β blockade would have been expected to increase vascular permeability and extravasation of our probe into tissues (hence decreasing the BPA) instead of increasing retention in blood. Moreover, we do not see significant changes in the levels of VEGF-A, an essential factor in the modulation of vascular permeability. Nonetheless, we decided to include analysis of BPA and the SUV_{mean} after BPA normalization in the supplementary material (Fig S3C). Interestingly, upon normalization for BPA, we still find an increase in SUV in gut and lymph nodes with the first treatment cycle. The limitations of the approach and images interpretation have been emphasized in the text (lines 183-189, 509-531 and 601-604).

iv. It is unclear how the regions of interest were defined especially for the gut analysis. One has to be careful to avoid spillover from other organs and radioactive stool will muddle the analysis. Axial images especially of the lymph node regions showing the ROIs would be helpful for review and add to the data pool.

We acknowledge that it would have been helpful to include more details regarding our contour strategy. After checking our contour strategy with [redacted], we confirmed that major organs and intestinal cavity (gut) ROIs were drawn appropriately. We also confirmed that the gut had no prominent or obvious intraluminal uptake (hence no notable stool signal). Lymph nodes and spleen were more problematic and were re-drawn based on CT (for the axillary LN, we used Axillary level 1 contour in TCF Van Heijst Phys. Med. Biol 2017). Images showing the contour strategy were included as Fig S2 and Movies 9 and 10).

v. Where does the whole-body increase (figure S2) in signal come in AC2, BC3 and AC3 if all of the primates are injected with nearly the same amount of activity for each scan? Again SUV total is not a typical measure used for image analysis.

Thank you for pointing this out. Upon re-analysis of the images, we also noticed that the last 2 scans of 08M171 were problematic. We had to start the study in 08M171 few weeks after the initial 7 macaques. Hence, the last 2 scans in this macaque were performed with a different batch of 7D3- probe than all the other scans (which were done all with the same batch). The particularly lower liver signal (Fig S1) and higher kidney signal at the BC3 and AC3 time points in 08M171 revealed that there may have been an issue with probe stability in the new batch used for these scans. Hence, in the revised manuscript, we decided to exclude the BC3 and AC3 scans for 08M171 from all the analysis. This, in turn, revealed that the increase in whole-body signal was an artifact due to this issue and was not present once the data were excluded. We mention the exclusion on line 167 and in the legend of Fig 2.

vi. Were other correlative studies performed on tissue viral load (e.g. lymph nodes, marrow), by biopsy, and uptake of the ^{64}Cu agent as a function of TGF β inhibition over the 4 cycles?

vii. The correlation of the viral load in colorectal biopsies and the SUV (Figure S3) is not very strong. This does not seem to match the image data. Also the description of the gut analysis: Gut abdomen ROI – organs is unorthodox and may not account for stool in the intestines. Also, the vRNA concentration is typically reported in the HIV literature on the log scale. All the data in S3 is linear and not large changes in concentration over the course of treatment.

We included correlative studies between LN SUV_{mean} and LN CA-vRNA with the caveat that at the end of cycle 1 the LN samples were inguinal and not axillary and that in some cases no vRNA was recovered from the sample. Interestingly, vRNA appears to correlate better with the absolute SUV_{mean} value (Fig S3B) than with the value normalized by blood pool (not shown). We also changed SUV Total to SUV mean in the correlation with gut (Fig S3A). Considering that gut biopsies are collected from random sites in tissues and do not represent repeated sampling at the same location, the correlation of gut SUV with gut vRNA appears quite good. Changes in cell-associated vRNA in a macaque under ART are not expected to be very large since they likely originate from a handful of latently reactivated cells.

viii. There is a disconnect between the later images for some of the primates showing increased activity throughout the body and the viral load in blood and tissues.

ix. The authors are strongly encouraged to have the PET imaging data thoroughly re-analyzed by a medical physicist or scientist with experience in PET image analysis and modeling.

In general the authors need to decide between keeping the imaging data in this manuscript, after considerable reanalysis, or remove it and publish with the remaining overall strong data. In the current state the imaging data does not strengthen the manuscript and may even detract from the strength of the biological data.

We thank the reviewer for the suggestion to consult with a nuclear medicine expert (particularly one with know n expertise in analysis of immunoPET/CT data such as [redacted]). This process led us to identify significant issues with our images that we initially overlooked. These included the problematic scans of the last 2 time points of 08M171 and the changes in blood pool activity that explain, at least in part, the large increases in signal

in the last 2-3 scans in all the animals. Unfortunately, although we are committed to finding the resources to image galunisertib-treated uninfected control animals, not even this control may ultimately be able to fully explain the increased activity that we suspect may have to do with modified pharmacokinetics of antigen-probe complexes. Nonetheless, the increased activity in gut and lymph nodes present in both normalized and non-normalized SUV constitutes important supporting data to our study. Moreover, it provides critical insight related to the potential impact of the drug on the pharmacokinetics of large molecules and complexes. Therefore, we think it should be included in the manuscript accompanied with a disclosure of all the limitations of this analysis for an appropriate interpretation of the data and conclusions. As noted above, we also hope to share the raw images with the scientific community in appropriate repository or upon demand for reanalysis to foster the debate.

Edits:

Page 4 line 103. Remove “s” from “T cells”

Done

Throughout manuscript and figures. ml should be mL

Done

Page 17 line 425 Remove “s” from “3 folds”

Done

Reviewer #3 (Remarks to the Author):

This manuscript by Kim et al represents an clear contribution to our knowledge of cytokines that influence the ability of HIV to remain latent in non-replicating T cell subsets. These studies in the SIV-macaque model represent an important step toward our ability to test TGF-beta blockade as a latency reversal agent in humans.

One area of concern is with regard to the ability of the authors to determine that it was an increase in effector or transitional effector T cells that was the major contributor in reversing the latency phenotype. Authors need to more clearly discuss the evidence for this in the results/discussion to make a compelling case that this is the key mechanism for the observed alterations in viral levels.

We thank the reviewer for raising this important point. We interpreted our data to indicate that the generation of transitional effector cells underlies latency reversal, but we have not proved this (we have now clarified this. See lines 606-608.) We found a direct association between the decrease in TCF1 and increase in effector function and virological variables indicating reactivation from latency. Without purposely interfering with these changes, we cannot prove the casual link between increase effector phenotype and latency reversal. More studies are needed to dissect the full mechanism explaining TGF- β -blockade driven latency reversal.

Also, authors need to more clearly discuss the use of the word ‘transitional’ in the title. The only place where they use this word in the results/discussion is on line 458, where they hypothesize that the cells are in the beginning of the effector stage. Outlining the evidence and a better definition for what they mean by transitional would be useful in understanding the relationship between the increased viral expression with this altered T cell phenotype (which appears to be primarily described as TCF1 low in the manuscript).

TGF- β impacts many cellular subsets and its effect is highly context dependent. Therefore TGF- β blockade is expected to have an heterogeneous effect on different subsets within both the T cell and myeloid cell compartments. As such we could not identify a single enriched population at the end of the treatment. However, the most prominent effect involved the generation of T cells that appeared to be transitioning to an effector phenotype. This conclusion is based on the following observations. TCF1 downregulation was the most profound

and consistent change throughout the different on-off cycles. Other notable changes at the protein level included an increase in CD95 and an initial decrease (shedding) of CD62L. However, at the same time, there were no change (or if anything a decrease) in tissue retention markers like CD69 or in markers of cellular proliferation such as Ki67. As such, it appears to us that galunisertib, rather than driving an increase in effectors, stimulates a transition within most T cells *toward* an effector phenotype. Supporting this interpretation, we observe basically no changes in exhaustion markers. On the transcriptional levels, changes are overall consistent with an increase in effector signature, increased transcriptional machinery and higher metabolic activity. This evidence summarizes the impact of the TGF- β blockade on the overall CD4 T cell population level (including effects that may be specific for naïve or Treg or other T cell subsets). This is further supported by the scRNAseq analysis where most of the changes were similar in different T subsets. Hence, we chose to use a terminology that highlights the temporal-dependency and major direction of the effect, “transitional effector”. We thank the reviewer for the help improving the manuscript. We articulated what we mean by “transitional” in the discussion (lines 583-590).

Minor issues:

Methods: concentrations of ART drugs needs to be added. Anti-PD1 antibody concentration utilized needs to be added.

Added.

Abbreviations: IPDA abbreviation not explained during first usage

Added

Figure 4G. top label is missing from some of the plots.

Adjusted.

4H and 4I: need to better explain how the populations were derived (supplemental figure) and to identify important populations in the figure legend.

Done

Line 311: there is no figure 6J

It was 6I. Corrected.

Figure 7. TNF alpha appears to be increased similarly irrespective of peptide, and in the absence of peptide, to a similar extent. Text should reflect that the T cells are more TNF-alpha producing in general. Unless there is a statistical difference between gag peptides and DMSO that is not stated or obvious. Current sentence is unclear in this regard (Lines 332-333).

The increase was similar, and the sentence was modified to improve clarity (lines 419-420).

Reviewers' Comments:

Reviewer #1:

Remarks to the Author:

The authors have now responded appropriately to concerns.

Reviewer #2:

Remarks to the Author:

The authors made significant revisions to the manuscript that has greatly improved the quality. The authors should be commended for their thorough and thoughtful responses to the review. There are a few minor comments that remain to be addressed.

1) The authors engaged [redacted] to review the analysis of the imaging data. The revisions made to the imaging data analysis and the corresponding sections in the manuscript have strengthened the interpretation and support the other findings in the study. One note- [redacted] should be given credit for his participation in the imaging data analysis either as a co-author or at a minimum he should be credited in the acknowledgements.

2) The authors did convert to SUVmean for the data presented in the main text. They did include the SUVtotal data in the supplementary data. The authors presented the rationale for using SUVtotal as this had been used in previous publications. They also described looking at %ID/organ data and found this to be very dependent on the region of interest (ROI) for the full organs. One may contend that SUVtotal is similar to %ID/organ and highly dependent on organ size (ROI). SUVtotal is not normalized to the size of the region like SUVmean. For this reason it is hard to interpret comparisons of SUV total from one primate to the next. If the tissue / ROIs are not the same size this can impact the SUVtotal value. Intra-primate temporal comparisons are likely to be suitable as the organ or tissue may not change size that rapidly. Therefore, in figure S3A the heavy black line (mean of the SUVtotal data) should be removed as this is not meaningful in this context.

3) Supplemental Figures 2 and 4: As the authors did in figure 2 please add a sentence indicating that no increase in uptake was seen for primate 08M171 and these data were eliminated from the analysis. Figure S4 appears to be incomplete. A (colorectal biopsies) and B (FNA) are defined but there is no A and B in the figure. Also the individual graphs are not labeled. These appear to be individual primate data but they are not labeled.

4) Figure 2 legend Line 1101 – change “are show” to “are shown”

Reviewer #3:

Remarks to the Author:

In the revised version of this manuscript my concerns have been addressed. The study represents an important contribution to the HIV-Cure field.

REVIEWERS' COMMENTS

Reviewer #1 (Remarks to the Author):

The authors have now responded appropriately to concerns.

Reviewer #2 (Remarks to the Author):

The authors made significant revisions to the manuscript that has greatly improved the quality. The authors should be commended for their thorough and thoughtful responses to the review. There are a few minor comments that remain to be addressed.

1) The authors engaged [redacted] to review the analysis of the imaging data. The revisions made to the imaging data analysis and the corresponding sections in the manuscript have strengthened the interpretation and support the other findings in the study. One note- [redacted] should be given credit for his participation in the imaging data analysis either as a co-author or at a minimum he should be credited in the acknowledgements.

We agree with the reviewer. However, we prefer to respect [redacted] wish not to be acknowledged in the manuscript.

2) The authors did convert to SUVmean for the data presented in the main text. They did include the SUVtotal data in the supplementary data. The authors presented the rationale for using SUVtotal as this had been used in previous publications. They also described looking at %ID/organ data and found this to be very dependent on the region of interest (ROI) for the full organs. One may contend that SUVtotal is similar to %ID/organ and highly dependent on organ size (ROI). SUVtotal is not normalized to the size of the region like SUVmean. For this reason it is hard to interpret comparisons of SUV total from one primate to the next. If the tissue / ROIs are not the same size this can impact the SUVtotal value. Intra-primate temporal comparisons are likely to be suitable as the organ or tissue may not change size that rapidly. Therefore, in figure S3A the heavy black line (mean of the SUVtotal data) should be removed as this is not meaningful in this context.

The line indicating the mean was removed.

3) Supplemental Figures 2 and 4: As the authors did in figure 2 please add a sentence indicating that no increase in uptake was seen for primate 08M171 and these data were eliminated from the analysis. Figure S4 appears to be incomplete. A (colorectal biopsies) and B (FNA) are defined but there is no A and B in the figure. Also the individual graphs are not labeled. These appear to be individual primate data but they are not labeled.

The sentence regarding 08M171 was added in the legends. Missing labels were added in Figure S4.

4) Figure 2 legend Line 1101 – change “are show” to “are shown”

Corrected

Reviewer #3 (Remarks to the Author):

In the revised version of this manuscript my concerns have been addressed.
The study represents an important contribution to the HIV-Cure field.